Corrected: Author correction

# HuR regulates telomerase activity through *TERC* methylation

Hao Tang[1,2], Hu Wang[3,4], Xiaolei Cheng[1], Xiuqin Fan[1], Fan Yang[3], Mengmeng Zhang[5], Yanlian Chen[6], Yuyang Tian[6], Cihang Liu[1], Dongxing Shao[1], Bin Jiang[1], Yali Dou[7], Yusheng Cong[3,4], Junyue Xing[1], Xiaotian Zhang[1], Xia Yi[1], Zhou Songyang[6], Wenbin Ma [6], Yong Zhao[6], Xian Wang[2], Jinbiao Ma [5], Myriam Gorospe[8], Zhenyu Ju[3,4] & Wengong Wang [1]

Telomerase consists of the catalytic protein TERT and the RNA *TERC*. Mutations in *TERC* are linked to human diseases, but the underlying mechanisms are poorly understood. Here we report that the RNA-binding protein HuR associates with *TERC* and promotes the assembly of the *TERC*/TERT complex by facilitating *TERC* C106 methylation. Dyskeratosis congenita (DC)-related *TERC* U100A mutation impair the association of HuR with *TERC*, thereby reducing C106 methylation. Two other *TERC* mutations linked to aplastic anemia and autosomal dominant DC, G107U, and GC107/108AG, likewise disrupt methylation at C106. Loss-of-HuR binding and hence lower *TERC* methylation leads to decreased telomerase activity and telomere shortening. Furthermore, HuR deficiency or mutation of *mTERC* HuR binding or methylation sites impair the renewal of mouse hematopoietic stem cells, recapitulating the bone marrow failure seen in DC. Collectively, our findings reveal a novel function of HuR, linking HuR to telomerase function and *TERC*-associated DC.

[1] Department of Biochemistry and Molecular Biology, Beijing Key Laboratory of Protein Posttranslational Modifications and Cell Function, School of Basic Medical Sciences, Peking University Health Science Center, 38 Xueyuan Road, Beijing 100191, China. [2] Department of Physiology and Pathophysiology, School of Basic Medical Sciences, Peking University Health Science Center, 38 Xueyuan Road, Beijing 100191, China. [3] Key Laboratory of Regenerative Medicine of Ministry of Education, Institute of Aging and Regenerative Medicine, Jinan University, Guangzhou 510632, China. [4] Institute of Aging Research, Hangzhou Normal University, School of Medicine, Hangzhou 311121, China. [5] Department of Biochemistry, School of life Sciences, Fudan University, 2005 Road Songhu, Shanghai 200433, China. [6] Key Laboratory of Gene Engineering of the Ministry of Education, State Key Laboratory of Biocontrol, School of Life Sciences, Sun Yat-sen University, Guangzhou 510006, China. [7] Department of Pathology and Biological Chemistry, University of Michigan, 1301 Catherine Street, Ann Arbor, MI 48105, USA. [8] Laboratory of Genetics and Genomics, National Institute on Aging, National Institutes of Health, 251 Bayview Blvd., Baltimore, MD 21224, USA. These authors contributed equally: Hao Tang, Hu Wang, Xiaolei Cheng, Xiuqin Fan, Fan Yang. Correspondence and requests for materials should be addressed to Z.J. (email: zhenyuju@163.com) or to W.W. (email: wwg@bjmu.edu.cn)

The majority of human cancer cells and germ line cells express telomerase, a ribonucleoprotein with reverse transcriptase activity that adds telomeric DNA repeats to the end of telomeres. The telomerase holoenzyme includes hTERT, the protein catalytic subunit, and *TERC*, a RNA subunit. The structural integrity of *TERC* is important for the assembly of the telomerase holoenzyme that regulates telomerase activity[1]. *TERC* serves as a template for telomerase to catalyze the addition of single-stranded telomere DNA repeats onto the 3′ ends of linear chromosomes[2,3]. Telomerase dysfunction caused by human *TERC* mutations is linked to numerous human diseases, including pulmonary fibrosis, human cancer, and premature aging syndromes, such as dyskeratosis congenita (DC) and aplastic anemia[1,4–6]. However, the mechanisms whereby these mutations cause telomerase dysfunction are largely unknown.

Methylation is a prevalent post-transcriptional modification for almost all RNA species[7–9]. RNA methylation is of critical importance for the translation[10], RNA stability and RNA processing[11,12]. Apart from tRNA, rRNA, and the mRNA 5′cap, small non-coding RNAs, such as piwi RNA, Drosophila siRNA, and microRNAs are also methylated[11]. Although, m6A is the predominant methylation site[13], m5C is also widely identified in human coding and non-coding RNAs[9,10]. Interestingly, m5C sites are also found in *TERC*[9]. However, the impact of m5C sites on telomerase activity has not been addressed.

Many *TERC*-associated proteins, such as DKC1 (dyskerin), TCAB1, the core components of box H/ACA small nucleolar ribonucleoprotein particles (snoRNPs), GAR1, NHP2, NOP10, Pontin/Reptin, DAXX, and telomeric proteins TIN2 and TPP1 were found to regulate telomerase activity[14–23]. HuR [human antigen R, also known as ELAVL1 (embryonic lethal abnormal vision-like 1)], the ubiquitously expressed member of Hu/ELAV RNA-binding protein family, has been described to regulate the post-transcriptional fate of a number of coding and non-coding RNAs[24–26], thereby regulating many cell activities (proliferation, survival, apoptosis, senescence, and differentiation) and affecting

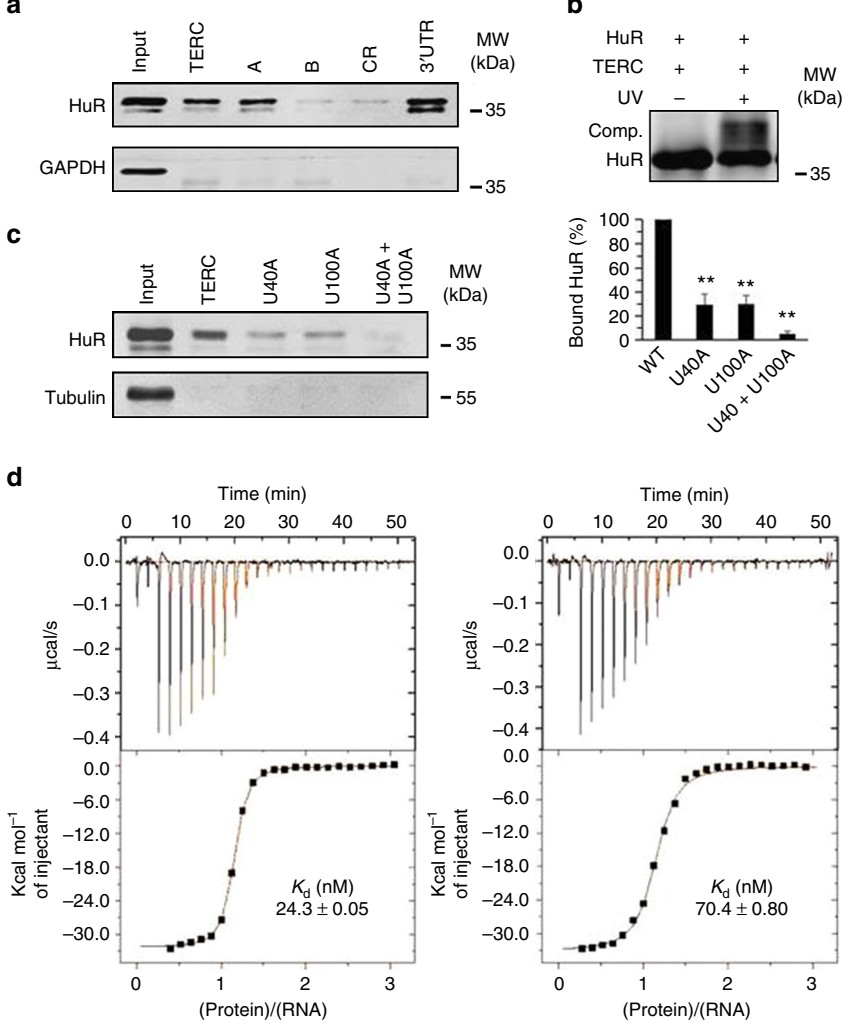

**Fig. 1** HuR interacts with *TERC* in vitro. **a** RNA pull-down assays were performed using HeLa cell lysates and in vitro-transcribed RNAs depicted in Supplementary Fig. 1a. The presence of HuR in the pull-down materials was assessed by western blot analysis. *p16* 3′-UTR and CR (coding region) served as positive and negative controls, respectively. A 5-µg aliquot input (Inp.) and binding to *GAPDH* RNA were also assessed. **b** Purified his-HuR and in vitro-transcribed *TERC* was used for UV-crosslinking rEMSA assays. The covalently bound HuR was detected by western blotting. **c** Left, the association of HuR with *TERC* variants bearing mutations U40A, U100A, or U40A + U100A (Supplementary Fig. 1b) was determined by using RNA pull-down assays, as described in Fig. 1a. Right, quantification of the bands on the western blot (left); data are the means ± SD of the signals from three independent experiments and significance was analyzed by Student's *t*-test (**$p < 0.01$). **d** In vitro ITC measurements of direct interactions between HuR and RNA oligonucleotides derived from human *TERC*, AUUUUUUGUCU (positions 37–47) and GUUUUUCUCG (positions 98–107), as described in Methods section. The dissociation constant ($K_d$) is calculated from 12 titrations and is indicated in the graphs

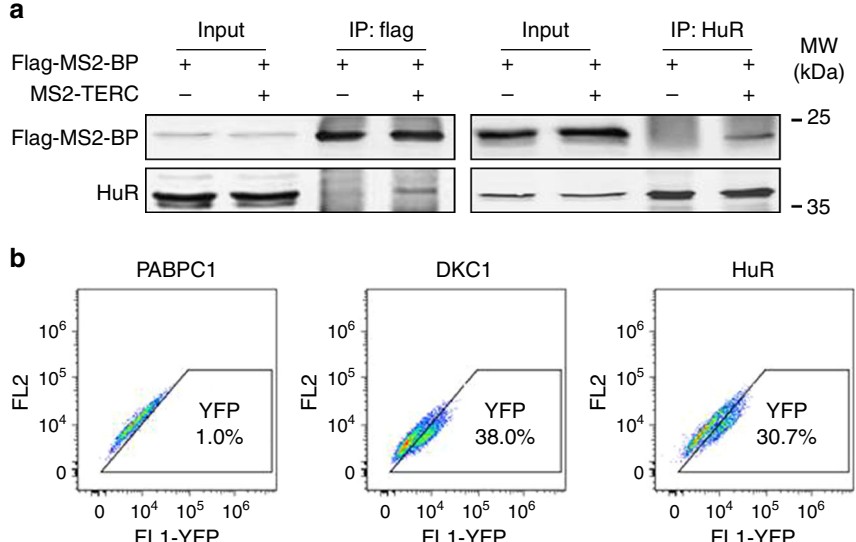

**Fig. 2** HuR interacts with *TERC* in cells. **a** U2OS cells were co-transfected with a vector expressing flag-MS2-BP together with a vector expressing MS2-*TERC* or an empty vector. Forty-eight hours later, lysates were prepared and subjected to IP assays to assess the association of HuR with RNA MS2-*TERC*. Data are representative from three independent experiments. **b** The TriFC system was used for detecting the interactions between HuR and *TERC* in cells by flow cytometry. DKC1 and PABPC1 served as positive and negative controls, respectively

broader processes, such as cancer and aging. Given that telomere metabolism is also involved in similar cellular and disease processes[27–29], we sought to elucidate the influence of HuR in telomere biology.

In this study, we describe evidence that HuR associates with *TERC* in vitro and in cultured cells. The association of HuR with *TERC* was required for the maintenance of *TERC* methylation and hence telomerase activity. Additionally, the regulation of telomerase activity by HuR was found to impact on the renewal of hematopoietic stem cells (HSCs) and was linked to dyskeratosis congenita, aplastic anemia, and autosomal dominant dyskeratosis congenita.

## Results

**HuR associates with *TERC* in vitro and in cells**. The association of HuR with *TERC* was studied by RNA pull-down assays using HeLa cell lysates and in vitro-transcribed, biotinylated *TERC* (full-length and fragments; Supplementary Fig. 1a). Western blot analysis revealed that HuR was presented in the complexes pulled down by using biotinylated full-length *TERC* and fragment A (positions 1–139), but not fragment B (positions 140–451) (Fig. 1a), suggesting that HuR was capable of associating with *TERC*. To test whether HuR bound *TERC* directly, recombinant, in vitro-purified his-HuR and in vitro-transcribed *TERC* were subjected to UV-crosslinking EMSA analysis (Methods section). As shown, a UV-crosslinked complex comprising purified his-tagged HuR and *TERC* was detected by western blot analysis (Fig. 1b), confirming the direct binding of HuR to *TERC*. According to the database of RNA-binding protein specificities (http://rbpdb.ccbr.utoronto.ca), the *TERC* RNA segments UUUUUU (positions 38–43) and GUUUUUC (positions 98–103) are potential sites for the binding of HuR. Therefore, further RNA pull-down assays were carried out by using *TERC* variants bearing mutations in UUUUUU, GUUUUUC, or both sites (Supplementary Fig. 1b). Mutating U40 or U100 residues (U40A or U100A) reduced greatly the association with HuR (by ~70.7% and ~70.4%, respectively; $p < 0.01$, Student's $t$-test), while mutating both (U40A + U100A) almost eliminated completely

this association (Fig. 1c). Therefore, *TERC* UUUUU and GUUUUUC are the major motifs for HuR binding. These results suggest that the association of HuR with *TERC* may be linked to DC, since U100A is a DC-related mutation[4]. Interestingly, UUUUU and GUUUUUC are conserved in mammals (Supplementary Fig. 1c), suggesting that the association of HuR with T*ERC* may be a common event in this class of vertebrates. By using isothermal titration calorimetry (ITC) assays, the dissociation constant ($K_d$) (fitting from 12 titrations) for UUUUUU and GUUUUUC was found to be ~24.3 ± 0.05 nM and ~70.4 ± 0.80 nM (± values denote standard errors), respectively (Fig. 1d, Supplementary Fig. 2 for negative control).

To test the interaction of *TERC* with HuR in cells, we employed human osteosarcoma U2OS cells, which do not express endogenous human TERT (hTERT) or *TERC*[30]. After transfecting U2OS cells with a vector that expressed a flag-tagged MS2-binding protein (flag-MS2-BP) alone or together with a vector that express tagged *TERC* (MS2-*TERC*), cell lysates were prepared and subjected to immunoprecipitation assays (IP) by using anti-HuR or anti-flag antibodies. The presence of flag-MS2-BP or HuR in the IP materials were assessed by western blot analysis. As shown, HuR associated with flag-MS2-BP in the presence, but not in the absence of MS2-*TERC* (Fig. 2a), indicating that HuR may associate with *TERC* in cells. We also tested the association of HuR with *TERC* in cells by using the TriFC system, which identifies protein-RNA interactions in cells[31]. As shown in Fig. 2b, HuR and DKC1 (positive control) associated with *TERC* in cells, while the nuclear protein PABPC1 (negative control) did not. In agreement with this finding, we observed the co-localization of HuR and *TERC* (Supplementary Fig. 3a), as well as the localization of HuR in Cajal bodies, where *TERC* undergoes maturation, as evidenced by the co-localization of HuR and the Cajal body marker Coilin (Supplementary Fig. 3b). In sum, HuR binds to *TERC* in vitro and in cells.

**HuR regulates telomerase activity**. Next, we tested whether HuR complexes with hTERT by associating with *TERC*. To this end, U2OS cells were transfected with a vector expressing flag-hTERT

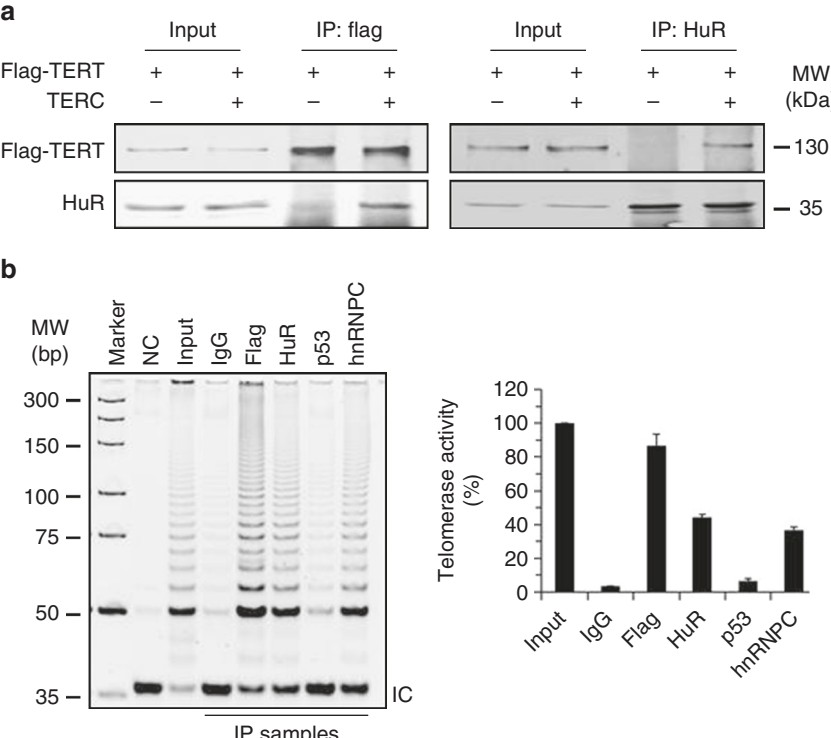

**Fig. 3** HuR associates with hTERT in a *TERC*-dependent manner. **a** U2OS cells were co-transfected with a vector expressing flag-hTERT together with a vector expressing *TERC* for 48 h. IP assays were performed to assess the association of flag-hTERT and HuR. Data are representative from three independent experiments. **b** HeLa cells were transfected using a vector that expressed flag-hTERT. Forty-eight hours later, lysates were prepared and subjected to IP assays by using antibodies indicated. The IP materials were used to test the telomerase activity by using TRAP assays. Data are the means ± SD from three independent experiments

alone or together with a vector expressing *TERC*. IP assays were performed by using the cell lysates and antibodies recognizing HuR or flag. As shown in Fig. 3a by western blot analysis, HuR associated with flag-hTERT in the presence, but not in the absence of *TERC*, indicating that HuR associates with hTERT in a *TERC*-dependent manner. We also tested the telomerase activity in the materials immunoprecipitated from human cervical carcinoma HeLa cells expressing flag-hTERT by using the telomere repeat amplification protocol (TRAP). As shown in Fig. 3b, telomerase activity was detected in the materials immunoprecipitated by using anti-HuR, anti-flag, or anti-hnRNPC (positive control) antibodies, but not in materials immunoprecipitated by using anti-IgG or anti-p53 antibodies (negative controls). These results further confirmed the association of HuR with hTERT.

To further analyze the influence of HuR on telomerase activity, HeLa cells were stably transfected with a vector expressing HuR shRNA. TRAP assays were performed to evaluate the influence of HuR silencing on telomerase activity. As shown, cells with silenced HuR exhibited much lower telomerase activity (<50% by 2, 30, and 60 days after silencing HuR) than control shRNA-transfected cells (Fig. 4a). Analysis of the telomere length in cells in which HuR was stably silenced, measured by southern blot analysis, revealed that HuR silencing caused a gradual shortening of telomeres (~3.44 kb vs. ~3.50 kb by day 2; ~3.5 kb vs. ~2.7 kb by day 30; ~3.5 kb vs. ~2.5 kb by day 60; Fig. 4b). In addition, modulating HuR levels did not alter the levels of flag-hTERT (expressed from full-length *hTERT* mRNA; Supplementary Fig. 4a), and neither transient nor long-term HuR silencing could affect *TERC* abundance (Supplementary Fig. 4b, c). These results indicate that HuR do not influence the abundance of hTERT or *TERC* but may instead regulate their

function. To test this possibility directly, IP analysis of UV-crosslinked RNPs was used to test the association flag-TERT and *TERC* in cells with silenced HuR. As shown, HuR knockdown reduced the association of *TERC* with hTERT by ~74.7% (p < 0.01, Student's t-test; Fig. 4c); the overall association of hTERT with *TERC* transcripts bearing mutations in the HuR-binding sites (Supplementary Fig. 1b) was markedly weaker (<50% lower) than the association of hTERT with wild-type *TERC* (Fig. 4d). These results indicate that HuR promotes the assembly of the *TERC*/hTERT core complex, in turn enabling telomerase activity.

**HuR regulates telomerase activity via *TERC* methylation**. In a previous study, C166, C323, and C445 were identified as m5C sites of *TERC*[9]. Among them, C323U is reported as a myelo-dysplastic syndrome-related mutation that reduces telomerase activity[32]. By using bisulfite RNA sequencing, we identified C106 as an additional m5C site in *TERC*. Mutation of C166 (C166U) or C445 (C445U) did not alter telomerase activity (Supplementary Fig. 5a-c). Knockdown of HuR reduced the methylation of C106 from ~32.6% to 10.1%, but did not reduce the methylation of C323 (Fig. 5a). Therefore, we focused on the role of C106 methylation in telomerase activity and the influence of HuR in this methylation event.

Mutation of HuR-recognizing sequences (U100A and U40A + U100A) reduced the methylation of C106 (with reduction from ~24.5% to 5.3% and 9.8%, respectively; p < 0.01, Student's t-test) (Fig. 5b), underscoring the importance of HuR in *TERC* C106 methylation. G107U and GC107/108AG are *TERC* mutations related to aplastic anemia and autosomal dominant DC[5,6]. Methylation of C106 was lost in G107U and GC107/108AG

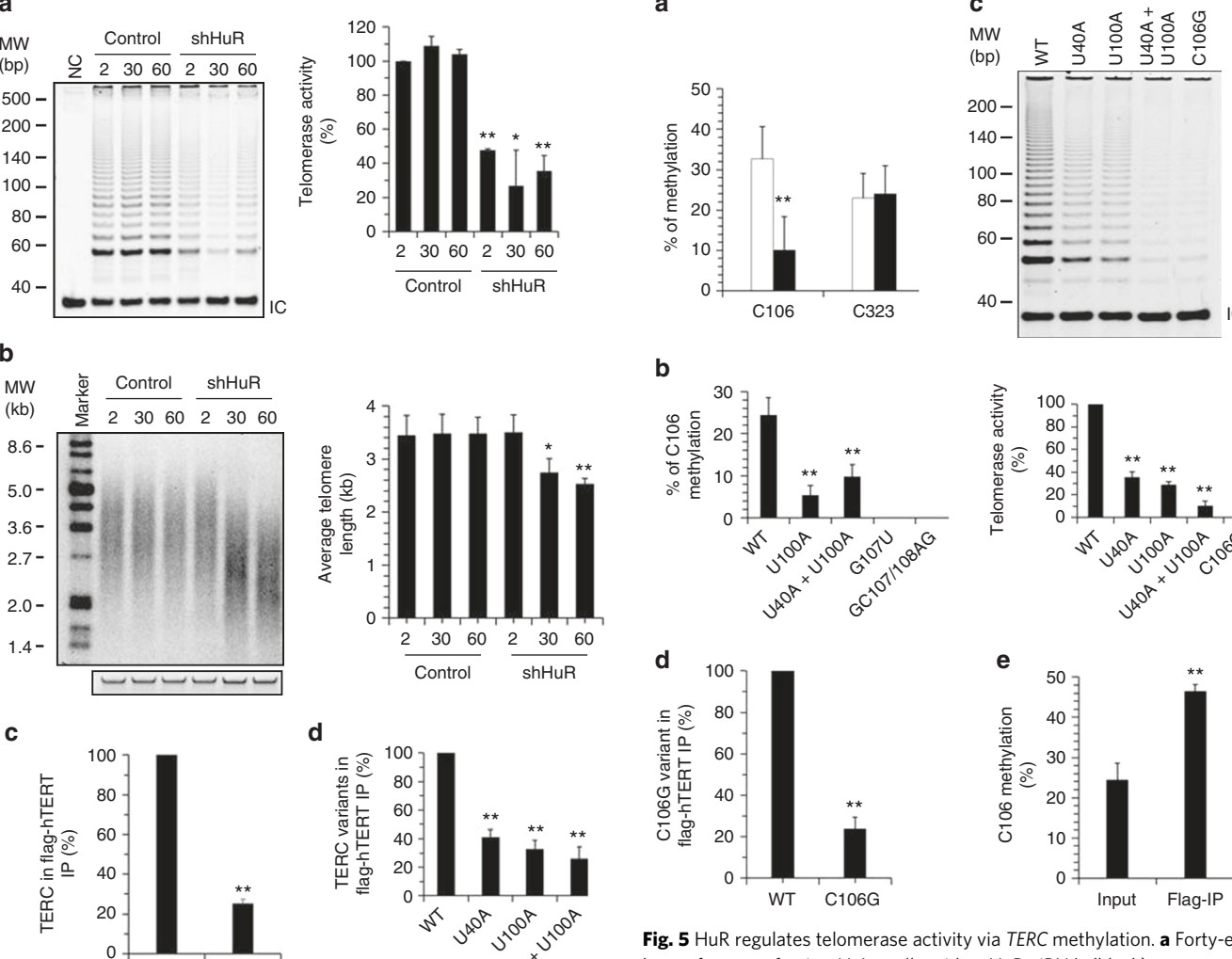

**Fig. 4** HuR regulates telomerase activity. **a**, **b** HeLa cells were infected with a lentivirus vector expressing shHuR at the times indicated. Telomerase activity (**a**) and telomere length (**b**) were analyzed by TRAP assays and southern blot analysis (**a** and **b**, left), respectively. The means ± SD from three independent experiments were analyzed for significance by Student's *t*-test (**a** and **b**, right) (*p < 0.05; **p < 0.01). **c**, **d** HeLa (**c**) or U2OS (**d**) cells were transfected with a vector expressing flag-TERT. Twenty-four hours later, cells were further transfected with a siRNA targeting HuR (**c**) or with a vector expressing *TERC* variants (**d**) (Supplementary Fig. 1b) and cultured for additional 48 h. UV-crosslinking followed by RNP IP assays were performed by using an anti-flag antibody. RNA isolated from IP materials was used for reverse transcription (RT) followed by real-time quantitative (q)PCR analysis to test the levels of flag-TERT-bound *TERC* (**c**) or its variants (**d**). Data in **d** were normalized against the levels of *TERC* or its variants. Data in **c** and **d** represent the means ± SD from three independent experiments; significance was analyzed by Student's *t*-test (**p < 0.01)

**Fig. 5** HuR regulates telomerase activity via *TERC* methylation. **a** Forty-eight hour after transfecting HeLa cells with a HuR siRNA (black) or a control siRNA (blank), RNA was isolated and used for bisulfite RNA sequencing analysis to measure the methylation of C106 and C323. **b** Forty-eight hour after transfecting HeLa cells with a vector expressing *TERC* variants or an empty vector (WT), RNA was isolated and used for bisulfite RNA sequencing analysis to measure C106 methylation in different variants (bearing point mutations were used for analysis). **c** U2OS cells were co-transfected with a vector expressing *TERC* variants C106G or U40A, U100A, or U40A + U100A, or C106G. Forty-eight hour later, TRAP assays were performed to determine the telomerase activity. **d** U2OS cells were co-transfected with a vector expressing flag-TERT together with a vector expressing *TERC* or its variant bearing C106G. UV crosslinking followed by RNP IP assays were performed to evaluate the association of flag-TERT with *TERC* and the variant bearing C106G. **e** HeLa cells were transfected with a vector expressing flag-hTERT or an empty vector (Input). Forty-eight hours later, lysates were prepared and subjected to UV crosslinking followed by IP assays using an anti-flag antibody. RNA prepared from the IP materials was further used for bisulfite RNA sequencing analysis to assess the methylation of C106. Data in **c**–**d** were normalized against the levels of *TERC* and its variants. Data in **a**, **b**, **c**, **d**, and **e** represent the means ± SD from three independent experiments; significance is analyzed by Student's *t*-test (**p < 0.01)

mutants (Fig. 5b), suggesting that the context of C106 was important for methylation. Similar to what was found for the HuR-binding motifs UUUUUU and GUUUUUC, C106 is also conserved in mammals (Supplementary Fig. 6). Moreover, an intact C106 site was important for telomerase activity, since mutation of C106 (C106G) greatly reduced telomerase activity (by ~77.1%; *p* < 0.01, Student's *t*-test) (Fig. 5c). Interestingly, mutation of HuR binding motifs (U40A, U100A, U40A + U100A) impaired telomerase activity by ~70 or more (Fig. 5c,

and Supplementary Fig. 7 for monitoring the levels of *TERC* variants). In keeping with earlier studies[5,6], mutations G107U and GC107/108AG similarly reduced telomerase activity without influencing the levels of these TERC variants (Supplementary Fig. 8a, b). C106G, G107U, or GC107/108AG mutants associated

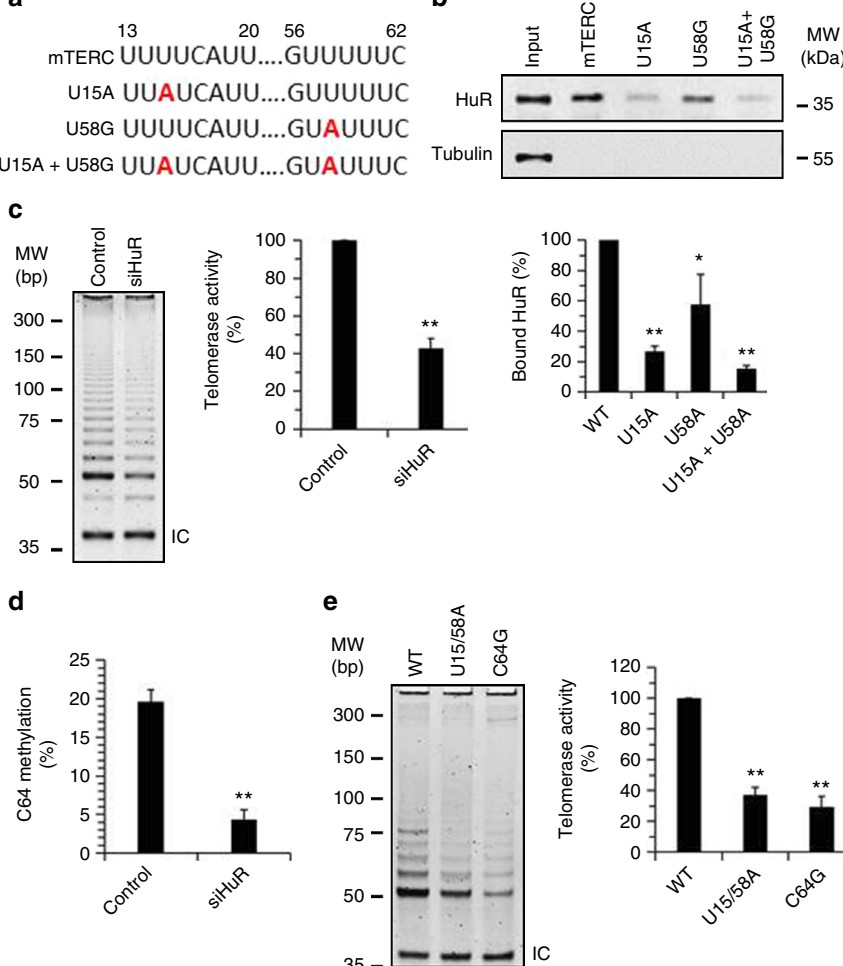

**Fig. 6** HuR regulates telomerase activity in mouse cells. **a** Schematic representation depicting the variants of *mTERC*. The point mutation sites are marked in red. **b** RNA pull-down assays were performed by using NIH3T3 cell lysates and in vitro-transcribed *TERC* variants depicted in Fig. 6a. The presence of HuR in the pull-down materials was assessed by western blot analysis. A 5-μg aliquot (input, Inp.) and proteins bound to GAPDH were included. Data represent the means ± SD of the band intensity from three independent experiments; significance was analyzed by Student's *t*-test (*$p < 0.05$; **$p < 0.01$). **c** NIH3T3 cells were transfected with a siRNA targeting HuR. Forty-eight hours later, TRAP assays were performed to assess the telomerase activity. **d** RNA prepared from cells described in Fig. 6c was subjected to bisulfite RNA sequencing to analyze the methylation of C64 in *mTERC*. **e** A vector expressing mouse TERT (mTERT) was used for expressing mTERT in rabbit reticulocyte in vitro translation system. In vitro-transcribed *mTERC* or its variant bearing U15A + U58A or C64G was added into the system and used for TRAP assays to assess the telomerase activity. Data in (**c**, **d**), and (**e**) represent the means ± SD from three independent experiments; significance was analyzed by using Student's *t*-test (**$p < 0.01$)

weakly with hTERT (Fig. 5d and Supplementary Fig. 8c), but not with HuR (Supplementary Fig. 8d, e and Supplementary Fig. 9). Therefore, the association of HuR with *TERC* enhances the methylation of C106, while methylation of C106 does not seem to influence the association of HuR with *TERC*.

To test the influence of C106 methylation on hTERT/*TERC* interaction, we measured the methylation of C106 in *TERC* immunoprecipitated from HeLa cells expressing flag-hTERT using an anti-flag antibody. As shown, the hTERT-bound *TERC* exhibited much higher C106 methylation (increasing from ~24.5% to 46.4%; Fig. 5e), suggesting that methylated *TERC* interacted more robustly with hTERT. To further test whether HuR-regulated *TERC* C106 methylation was important for maintaining telomerase length, HeLa cells were stably transfected with a vector expressing flag-hTERT alone or together with a vector expressing *TERC* or its variants (U40A + U100A or C106G). Telomere length was measured by southern blot analysis. As shown, co-expression of flag-hTERT with *TERC*,

but not with U40A + U100A or C106G mutants, extended markedly telomere length in HeLa cells ($p < 0.01$, Student's *t*-test) (Supplementary Fig. 10a-c). Together, HuR was found to enhance the methylation of *TERC*, in turn promoting the assembly of the *TERC*/hTERT complex and elevating telomerase activity; *TERC*-bearing mutations linked to DC, aplastic anemia, or autosomal dominant DC disabled *TERC* methylation (U100A, G107U, and GC107/108AG) or its association with HuR (U100A).

**HuR-telomerase axis impacts on the renewal of mHSCs.** Bone marrow failure is the leading cause of death in DC patients[1]. Therefore, we asked whether the HuR-telomerase axis influenced the renewal of mHSCs. To this end, we first addressed whether the impact of HuR and *TERC* methylation in telomerase activity observed in human cells also occurred in mouse cells. As shown, mutation of mouse *TERC* (*mTERC*) HuR-binding sites prevented

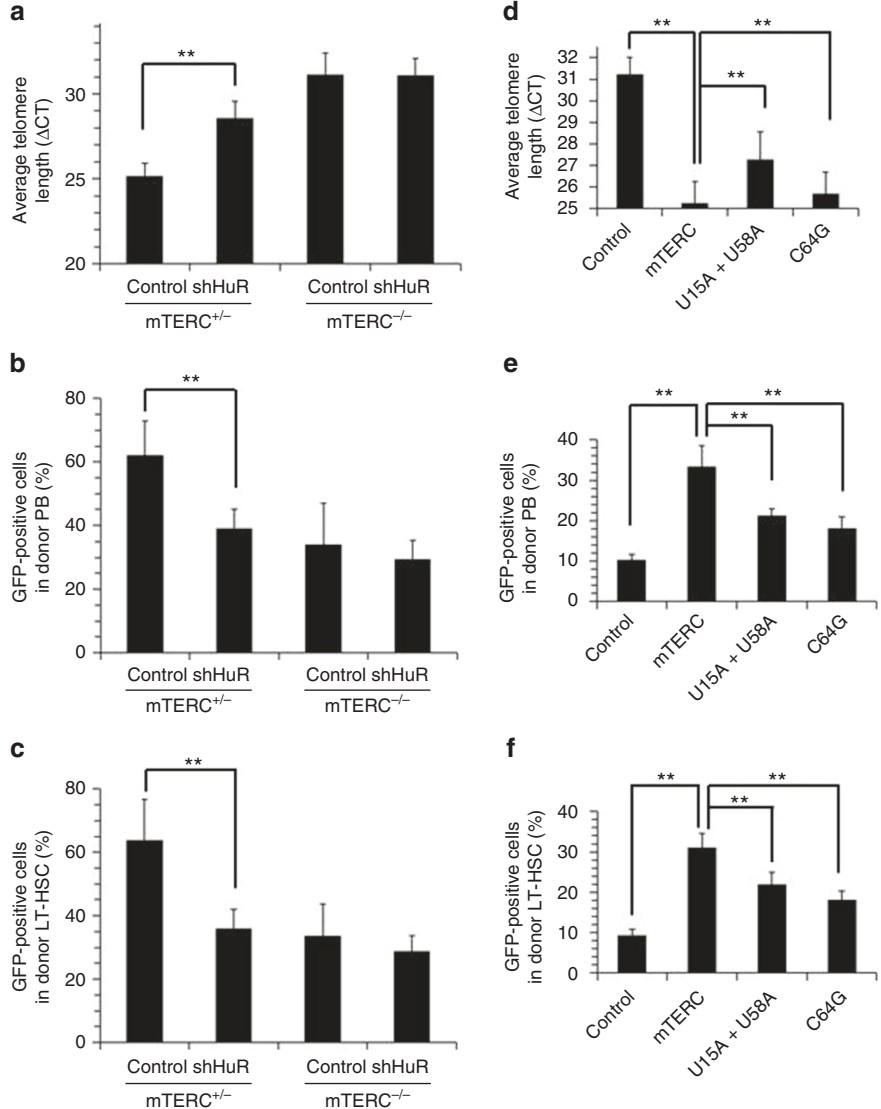

**Fig. 7** HuR-telomerase axis impacts on the renewal of mHSCs. **a** Twelve weeks after transplanting, GFP⁺ donor-derived LT-HSCs (Lin⁻Sca1⁺cKit⁺FLT3⁺CD34⁻) were isolated from primary recipients of *TERC*⁺/⁻ group or G3*TERC*⁻/⁻ group (*n* = 5) and subjected to single-cell qPCR analysis to determine the telomere length. The ΔCT values of the qPCR data are represented as the means ± SD; significance was analyzed by using Student's *t*-test (**p < 0.01). **b**, **c** Contribution of the GFP⁺CD45.2⁺ (with silenced HuR) cells to the indicated PB (**b**) and LT-HSC populations (**c**) at 12 weeks after transplantation, respectively. Data represent the means ± SD from five mice; significance was analyzed by using Student's *t*-test (**p < 0.01). **d-f** G3*TERC*⁻/⁻ mHSC (LSK) cells were infected with lentiviruses expressing GFP together with *TERC* or its variants (U15A + U58A or C64G). The sorted cells were further treated and analyzed same as described in (**a-c**)

HuR binding (Fig. 6a, b). Knockdown of HuR reduced telomerase activity and m*TERC* methylation at C64, the conserved m5C site (Fig. 6c, d). Mutations of the m*TERC* sequences that HuR binds (U15A + U58A) or the m5C site (C64G) impaired telomerase activity (Fig. 6e). Therefore, the regulation of telomerase activity regulating by HuR and *TERC* methylation appears conserved in mouse cells.

Next, we tested whether HuR-telomerase regulatory process impacts on hematopoietic stem cell (HSC) function. To this end, Lineage⁻Sca1⁺cKit⁺ (LSK) from m*TERC*⁺/⁻ or G3 m*TERC*⁻/⁻ mice were infected with lentivirus expressing a shHuR or a control shRNA together with GFP. The GFP-positive cells were sorted and transplanted into recipient mice (*n* = 5). The transduction efficiency and FACS analysis for sorting efficiency are shown (Supplementary Fig. 11-13). As shown, knockdown of HuR shortened telomere length and attenuated the function of m*TERC*⁺/⁻ HSCs (*p* < 0.01, Student's *t*-test), but not

m*TERC*⁻/⁻ HSCs (Fig. 7a–c). These results support the notion that HuR regulates HSC telomere length and function in a *TERC*-dependent manner. We then constructed lentiviral vectors expressing wild-type m*TERC* or m*TERC* bearing mutations in the motif recognized by HuR (U15A + U58A or m5C site C64G). HSCs were then infected with the vectors and transplanted into m*TERC*⁻/⁻ mice. Strikingly, overexpression of wild-type m*TERC* effectively increased telomere length and rescued the impairment of m*TERC*⁻/⁻ HSC function, while overexpression of mutant m*TERC* was much less effective in restoring these phenotypes (Fig. 7d–f). Together, these results indicate that the regulation of telomerase activity by HuR influence mHSC function in a *TERC*-dependent manner.

## Discussion
The present study provides evidence that HuR regulates telomerase activity via *TERC* methylation in human cell lines and in

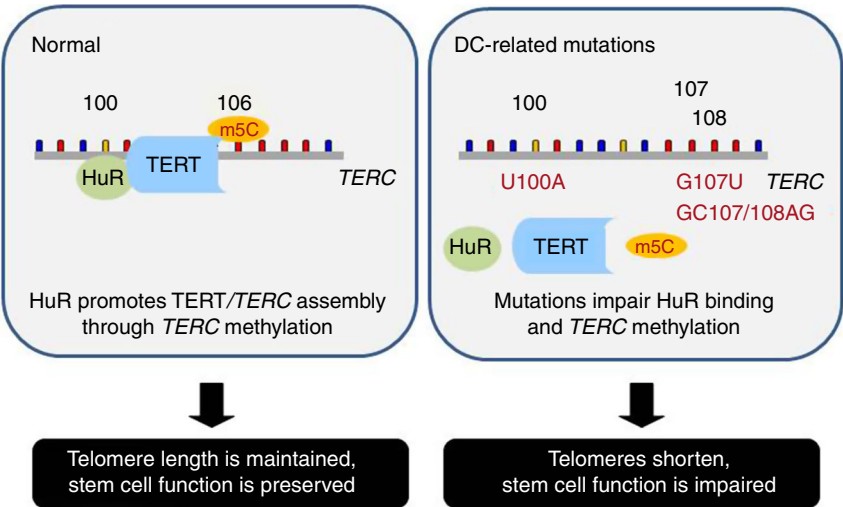

**Fig. 8** A model summarizes the findings in this study. Association of HuR with *TERC* promotes TERT/*TERC* assembly through enhancing the methylation of *TERC* (m5C) in C106. The reported DC-related mutations impair HuR binding (U100A) or the methylation of *TERC* (U100A, G107U, and GC107/108AG)

transplanted mHSCs (Fig. 8). In previous studies, HuR was shown to promote mouse progenitor cell survival by inhibiting the MDM2-TP53 axis[33]. Since telomere shortening activates TP53[34], the silencing of HuR and ensuing reduction in telomere length may have further increased TP53 levels. The conservation of HuR-interacting *TERC* region and the m5C site (Supplementary Fig. 1c and Supplementary Fig. 6) suggests that the HuR-telomerase regulatory paradigm may be conserved in mammals. Given that m5C methylation and HuR-RNA interactions occur ubiquitously in human cells[9,10,35], the influence of HuR in RNA methylation is unlikely to be limited to *TERC*. The finding that knockdown of HuR reduces the formation of double-stranded *TERC* RNA (Supplementary Fig. 14) suggests that the association of HuR with *TERC* may enhance *TERC* C106 methylation by altering the secondary structure of *TERC*; the influence of *TERC* methylation on the secondary structure of *TERC* remains to be further studied.

Apart from C106, cytosines C166, C323, and C455 have also been identified as m5C sites in *TERC*[9]. However, methylation at C166 and C455 did not appear to affect telomerase activity, since it remained unchanged after mutating C166 and C455 (Supplementary Fig. 5). In addition, although methylation at C323 may be important for maintaining telomerase activity[32], this modification was not modulated by HuR (Fig. 5a). Therefore, the various *TERC* methylation sites, the factors that mediate *TERC* methylation, and its impact on telomerase activity are complex. In addition, while the transplantation experiment in mouse HSCs are evidence of the impact of HuR-regulated *TERC* methylation in bone marrow failure syndromes (Fig. 7), the RNA methyltransferase catalyzing the C106 methylation and the reader of C106 methylation await identification.

The finding that DC-related mutations affect the association of HuR with *TERC* and C106 methylation set the stage for important future studies on the clinical impact of these findings on human hematopoietic stem cell homeostasis. Given that both HuR and telomerase promote human cancer and cell growth, and that they both inhibit aging, cell senescence, and apoptosis[24–29,36,37], we propose that the HuR-telomerase regulatory paradigm is a major component of these processes.

## Methods
**Cell culture, transfections, and knockdown of HuR**. Human osteosarcoma U2OS cells, human cervical carcinoma HeLa cells, and mouse NIH3T3 fibroblasts from

ATCC were cultured in Dulbecco's modified Eagle's medium (Invitrogen) supplemented with 10% fetal bovine serum, 100 units/ml penicillin, and 100 μg/ml streptomycin, at 37 °C in 5% $CO_2$. All plasmid transfections were performed using Lipofectamine 3000 (Invitrogen) following the manufacturer's instructions. To silence the expression of HuR transiently in both human and mouse cells, a siRNA targeting HuR (AAGAGGCAAUUACCAGUUUCA) or a control siRNA (UUGUUCGAACGUGUCACGUTT) was transfected using Oligofectamine (Invitrogen) following the manufacturer's instructions. Unless otherwise indicated, cells were collected for analysis 48 h after transfection.

**In situ hybridization and immunofluorescence analysis**. Cells were fixed with 4% paraformaldehyde for 30 min at room temperature and then permeabilized in 0.1% Triton X-100. For RNA FISH, cells were blocked with pre-hybridization buffer at 3 °C for 1 h and incubated in hybridization buffer (50% deionized formamide, 10% dextran sulfate, 2 mM vanadyl ribonucleotide complex, 0.002 mg/mL nuclease-free bovine serum albumin, 1 mg/mL *E. coli* tRNA and 250 μg/mL of N-50 DNA) containing *TERC* probes (fluorescein isothiocyanate-conjugated DNA probes complementary to human *TERC*) (TAKARA Bio Inc) at 37 °C overnight in a humidified chamber. Samples were washed three times with 2× SSC and then five times with 1× PBS before combining with immunofluorescence. For immunofluorescence, primary monoclonal antibodies recognizing coilin (Abcam, UK) or HuR (Santa Cruz) were used. Cy5 and/or FITC-conjugated secondary antibodies (Multiscience, China) were used for detection. Cells were mounted with DAPI. Fluorescence was detected and imaged using a Nikon Ti microscope.

**Western blot analysis**. Western blot analysis was performed following standard procedures. Cell lysates were size fractionated by sodium dodecyl sulfate-polyacrylamide gel electrophoresis (SDS-PAGE) and transferred onto nitrocellulose membranes. Monoclonal antibodies recognizing GAPDH (proteintech, Cat# 10494-1-AP (50 μg/150 μl), 1:2000), Tubulin (proteintech, Cat# 11224-1-AP (43 μg/150 μl), 1:2000), HuR (Santa Cruz, Cat#SC-5261 (200 μg/ml),1:1000), or anti-flag M2 antibody (Sigma, Cat# F1802 (1 mg/ml),1:2000) were used for the respective western blot analysis. Following incubation with the appropriate secondary antibody, signals were detected by using Odyssey CLx western blot detection system. All of the uncropped western blots with molecular weight indicated were included in Supplementary Figs. 15-24.

**Reverse transcription (RT) and real-time qPCR analysis**. For reverse transcription (RT) followed by real-time quantitative (q)PCR analysis to detect *TERC*, m*TERC*, and their variants, total cellular RNA was prepared using the RNeasy Mini Kit (QIAGEN) following the manufacturer's protocol, a reverse transcription (RT) reaction was performed by using primer GCGTCTCAACTGGTGTCGTG-GAGTCGGCAATTCAGTTGAGACGCGCATGTGTGAG, and the products from RT reaction were subjected to real-time qPCR analysis by using primer pairs TTCAGGCCTTTCAGGCCGCAGGAA and TGGTGTCGTGGAGTCGGC for *TERC*, and primer pairs ACCTAACCCTGATTTTCATTAGC and GGTTGTGA GAACCGAGTTCC for m*TERC*. U6 was used for normalization in all real-time qPCR assays. Primer GCGTCTCAACTGGTGTCGTGGAGTCGGCAATTCAGT TGAGACGCAAAATATGGAA was used for the reverse transcription of *U6*. Primer pairs CGCAAGGATGACACGCAAATTCG and TGGTGTCGTGG

AGTCGGC were used for real-time qPCR analysis of *U6*. The reverse primers used for real-time qPCR analysis of *TERC* and *U6* matched the sequence of the primers used for reverse transcription.

Details of the real-time qPCR analysis are as follows:

1. Add into sterile, nuclease-free tube on ice in the indicated order (12 μl in total): nuclease-free water, DNase-treated total RNA (1 μg), stem-loop gene-specific RT primer (20 pmol).
2. Mix gently, centrifuge briefly and incubate with the following program in a thermal cycler: 65 °C (5 min), 45 °C (5 min), 25 °C (5 min), 4 °C (hold).
3. Add the following components in the indicated order: 5× reaction buffer (4 μl), RNase inhibitor (1 μl; 20 U, Thermo Scientific, EO0381), dNTP Mix, 10 mM each: 2 μl (1 mM final concentration, Thermo Scientific, R0191), reverse transcriptase: 1 μl (200 U, Thermo Scientific, EP0441). Mix gently and centrifuge briefly.
4. Using a thermal cycler, proceed reverse transcription with the following program: 25 °C (10 min), 42 °C (60 min), 70 °C (10 min), 4 °C (hold).
5. Perform qPCR assays in the following programs (Biorad CFX96 real-time system): pre-denature at 95 °C for 10 min, use [95 °C (15 s), 60 °C (45 s)] for 40 cycles, then perform melting curve analysis and normalize using comparative Ct ($\triangle\triangle$Ct) quantification method.

**RNA pull-down and immunoprecipitation (IP).** For biotin pull-down assays, PCR-amplified DNA was used as template to transcribe biotinylated RNA by using T7 RNA polymerase in the presence of biotin-UTP. One microgram of purified biotinylated transcripts was incubated with 100 μg of cytoplasmic extracts for 30 min at room temperature. Complexes were isolated with paramagnetic streptavidin-conjugated Dynabeads (Dynal, Oslo), and the pull-down material was analyzed by western blotting.

For immunoprecipitation (IP) assays, whole-cell lysates were prepared by adding 200 μl of IP buffer (10 mM Hepes, pH 7.4, 50 mM β-glycerophosphate, 1% Triton X-100, 10% glycerol, 2 mM EDTA, 2 mM EGTA, 10 mM NaF, 1 mM DTT, protease Inhibitor Cocktail). IP reactions were carried out by using 20 μl of the resulting supernatant, diluting it with 1 ml of IP buffer, and adding 2 μg of the antibodies indicated above. The washes were performed as follows: three times with IP buffer, four times with a high-stringency buffer (100 mM Tris-HCl, pH 7.4, 500 mM LiCl, 0.1% Triton X-100, 1 mM DTT, protease Inhibitor Cocktail), and then three times with IP buffer.

**RNP IP assays.** For RNP IP (UV-crosslinking RNP IP) assays, cells were exposed to UVC (400 mJ/cm$^2$) and lysates were prepared for immunoprecipitation using monoclonal anti-HuR or anti-flag antibodies. The IP materials were washed twice with stringent buffer (100 mM Tris-HCl, pH 7.4, 500 mM LiCl, 0.1% Triton X-100, 1 mM DTT, protease Inhibitor Cocktail) and twice with the IP buffer. The RNA in RNP IP was assessed by RT-qPCR analysis.

**UV crosslinking RNA electrophoretic mobility shift assay.** For UV crosslinking RNA electrophoretic mobility shift assay (UV-crosslinking rEMSA), the RNA–protein interaction mixtures (0.02 mL) contained 50 mM Tris (pH 7.0), 150 mM NaCl, 0.25 mg/ml tRNA, 0.025 mg/ml bovine serum albumin, 500 nM of purified his-HuR and 500 nM of in vitro-transcribed *TERC*. Mixtures were incubated at 25 °C for 30 min, and digested with RNase T1 (100 U/reaction) for 15 min at 37 °C. After crosslinking of complexes through delivery of 1800 J/m$^2$, reactions were subjected to western blot analysis.

**Constructs.** For construction of the pcDNA 3.0 (+) vectors (Invitrogen) expressing *TERC*, *mTERC*, and the variants bearing *TERC* and *mTERC* mutations, primer pairs CGGGATCCGGGTTGCGGAGGGTGGGCCT and CGGAATTCGCATGT GTGAGCCGAGTCCTGG for human *TERC* (*TERC*), and CGGGATCCCACCTAA CCCTGATTTTCATT and CGGAATTCGGTTGTGAGAACCGAGTTCC for mouse *TERC* (*mTERC*) were used to amplify the full length of *TERC* and *mTERC*. These products were inserted between the *BamH I* and *EcoR I* sites of pcDNA 3.0 (+) vector. To construct pcDNA 3.0 (+) vectors expressing variants of *TERC* (U40A, U100A, U40A + U100A, C106G, G107U, GC107/108AG, C166U, C323U, and C445U) and *mTERC* (U15A, U58A, U15A + U58A, and C64G), we used circular site-directed mutagenesis technology. The primer pairs used are as follows: GGGGTGGTGGCCATTGTTTGTCTAAC and CAATGGCCACCACCCCTCCC AGGCCC for U40A, CTCCCCGCGCGCTGTATTTCTCGCTG and TACAGCG CGCGGGGAGCAAAAGCACG for U100A, GCGCGCTGTTTTTCTGGCTGAC TTTC and CAGAAAAACAGCGCGCGGGGAGCAAA for C106G, CGCGCTGT TTTTCTCTCTGACTTTCA and AGAGAAAAACAGCGCGCGGGGAGCAA for G107U, CGCGCTGTTTTTCTCAGTGACTTTCAG and CTGAGAAAAACAGC GCGCGGGGAGCAA for GC107/108AG, GTTCATTCTAGAGTAAACAAAAAA TGTCAG and TTGTTTACTCTAGAATGAACGGTGGAAGGC for C166U, GGG CTCTGTCAGGCCGTGGGTCTCTCG and ACGGCTGACAGAGCCCAACTCT TCGC for C323U, CCAGGACTCGGCTCATACATGCAATT and ATGAGCCG AGTCCTGGGTGCACGTCC for C445U, CGGGATCCCACCTAACCCTGATTAT CATT and CGGAATTCGGTTGTGAGAACCGAGTTCC for *mTERC* U15A, TTC TCCGCCCGCTGTGTTTCTCGCTG and CACAGCGGGCGGAGAACAAAGA

CCA for *mTERC* U58A, and GCCCGCTGTTTTTCTGGCTGACTTCC and CAG AAAAACAGCGGGCGGAGAACAAA for *mTERC* C64G. The pcDNA 3.0 (+) vector expressing U40A + U100A of *TERC* or U15 + U58A of *mTERC* was generated by mutating U100A of *TERC* or U58 of *mTERC* from template vector bearing U40A or U15A by using circular site-directed mutagenesis technology. To construct pcDNA 3.0 (+) vector (Invitrogen) expressing flag-tagged hTERT (human TERT), the full length of hTERT cDNA was amplified by using primer pairs ATAAGAATGCGGCCGCTATGCCGCGCGCTCCCCGCTGCCGAG and TGCTCTAGATCAGTCCAGGATGGTCTTGAAGTCT and inserted between the *Not I* and *Xba I* sites. To construct 3.0 (+) vector expressing flag-tagged mTERT (mouse TERT), primer pairs CAAGCTTATGACCCGCGCTCCTCGTTG and GGAATTCTTAGTCCAAAATGGTCTGAA were used to amplify the *mTERT* cDNA and inserted between the *Hind III* and *EcoRI* sites. 3× flag sequences (AGCTTGTCATCGTCATCCTTGTAATCGATGTCATGATCTTTATAATCACC GTCATGGTCTTTGTAGTCCATA) was synthesized and sub-cloned into the *Hind III* site.

To construct pcDNA 3.1 vector (Clontech) expressing flag-HuR, the full-length coding region of HuR was amplified by PCR using primer pairs GGAATTCATGGACTACAAGGACGACGATGACAAGTCTAATGGTTATGAA and GCTCTAGATTATTTGTGGGACTTGTTGG and inserted between the *EcoR I* and *Xba lI* sites of the pcDNA 3.1 vector.

To construct a vector expressing his-HuR in E.coli, the full-length coding region of HuR was amplified by using primer pairs CGGAATTCATGTCTAATGGTTAT GAAGA and CCCAAGCTTTTATTTGTGGGACTTGTTGG and inserted between the *EcoR I* and *Hind III* sites of The PET-28A (+) vector (QIAGEN).

To construct lentivirus vector (pHBLV-CMVIE-IRES-Puro) (HANBIO, China) expressing flag-tagged hTERT and HuR shRNA, hTERT cDNA were amplified by PCR by using primer pairs CGGAATTCATGGACTACAAAGACCATGACGGTG and GGAAGATCTTCAGTCCAGGATGGTCTTGAAGTCTG and inserted between the *EcoR I* and *Bgl II* sites. To construct lentivirus vector expressing HuR shRNA, synthesized oligomer pairs GATCCGAAGAGGCAATTACCAGTTTCAT TCAAGAGATGAAACTGGTAATTGCCTCTTCTTTTTTC and AATTGAAAAG AAGAAGAGGCAATTACCAGTTTCATCTCTTGAATGAAACTGGTAATTGC CTCTTCG were annealed and inserted between the *EcoRI* and *Bgl II* sites of pHBLV-U6-Puro vector (HANBIO, China).

To construct the Lentivirus vector SF-LV-EGFP (from Karl Lenhard Rudolph's lab, Leibniz Institute on Aging - Fritz Lipmann Institute, Jena, Germany) expressing *mTERC* and its variants, the *mTERC* fragments were amplified by PCR from the pcDNA 3.0 (+) vectors expressing *mTERC* and its variants by using primer pairs CCGCTCGAGACCTAACCCTGATTTTCATT and CGGAATTCGG TTGTG AGAACCGAGTTCC and inserted between the *XhoI* and *EcoR I* sites of SF-LV-EGFP vector. To construct vector expressing SF-LV-EGFP vector expressing mouse shHuR, oligomers TGCTGTTGACAGTGAGCGAAAGAGGCA ATTACCAGTTTCATAGTGAAGC ACAGATGTATGAAACTGGTAATTGCCT CTTCTGCCTACTGCCTCGGA and ACAGTGAGCGAAAGAGGCAATTACCA GTTTCATAGTGAAGCCACAGATGTATGAAACTGGTAATTGCCTCTTCTGC CTA were annealed and inserted between the *Xho I* and *EcoR I* sites of the SF-LV-EGFP vector.

To construct pSL-MS2 vector expressing *MS2-TERC*, primer pairs CCCAAG CTTGGGTTGCGGAGGGTGGGCCT and GGAATTCGCATGTGTGAGCCGAG TCCTGG were used to amplify *TERC* and inserted between the *Hind III* and *EcoR I* sites of the pSL-MS2 vector. To construct pCMV-7.1-3× flag (Sigma) expressing MS2-binding protein (MS2-BP), the fragment encoding coat MS2-BP was amplified by PCR by using primer pairs ATAAGAATGCGGCCGCTATGGGCT ACCCCTACGACGTG and TGCTCTAGATCAGGTGGCGACCGGTGGATCCG from pMS2-BP-YFP vector and inserted between the *Not I* and *Xba I* sites of the pCMV-7.1-3× flag vector.

**Preparation of the transcripts.** cDNA was used as a template to amplify the different fragments of *TERC* or *mTERC*. All 5′ ends contained the T7 promoter sequence 5′-CCAAGCTTCTAATACGACTCACTATAGGGAGA-3′ (T7). To amplify templates for synthesizing RNAs *TERC*, *TERC-A*, *TERC-B*, *mTERC*, *p16* CR (coding region), and p16 3′-UTR, we used following primer pairs; (T7) GGGTTGCGGAGGGTGGGCCT and GCATGTGTGAGCCGAGTCCTGG for *TERC*, (T7) GGGTTGCGGAGGGTGGGCCT and AGGCCGAGGCTTTTCCGCC for *TERC-A*, (T7) GCCGCCTTCCACCGTTCATT and GCATGTGTGAGCCGAG TCCTGG for *TERC-B*, (T7) ACCTAACCCTGATTTTCATTAGC and GGTTGT GAGAACCGAGTTCC for *mTERC*, (T7) AGCAGCATGGAGCCTTCG and GGTTCTTTCAATCGGGGAT for *p16* CR, and (T7) CGATTGAAAGAACCAG AGAG and GTTCTGCCATTTGCTAGCAG for *p16* 3′-UTR. The templates for variants of *TERC* (U40A, U100A, U40A + U100A, C106G, G107U, and GC107/ 108AG) and *mTERC* (U15A, U58A, U15A + U58A, and C64G) were amplified from the pcDNA 3.0 (+) vector expressing these variants (as described in 'Constructs') directly by using primer pairs for wild-type *TERC* and *mTERC*. PCR-amplified DNA was used as the template to transcribe biotinylated RNA by using T7 RNA polymerase in the presence of biotin-UTP.

**TriFC assays.** The vectors pBabe-cmv-NLS-HA-tdMCP-NYFP, plko-*TERC*-24Xms2-BSD, and FG-EH-DEST2-YFPc-ppw were used in TriFC system. To construct pBabe-cmv-NLS-HA-tdMCP-NYFP containing a nuclear localization

signal (NLS) and a hemagglutinin (HA) tag, two coat tdMCPs sequence linked by sequence ATCTACGCCATGGCTTCT were first cloned into a Gateway™ Entry vector then inserted into att R sites of pBabe-cmv-NYFP vector by Gateway LR reaction. To construct plko-TERC-24Xms2-BSD, we modified the pLKO.1-puro lentiviral backbone to replace the puromycin (puro) resistant selection with blasticidin (BSD) and mutated the EcoR I site GAATTC to GAATCC. The TERC and MS2 (24×) sequence linked by a Bam H I site was amplified by PCR and inserted between the EcoR I and Age I sites of plko-BSD lentiviral backbone vector. The pXPA-2/pMD2G or PCGP/VSVG packaging plasmids were packaged in 293T cells. HTC75 cells were infected with these viruses and the stably infected cells were sorted via screening with puro, BSD or G418 48 h later. The ternary cell line was analyzed by flow cytometry.

**Isothermal titration calorimetry (ITC) assays.** HuR constructs were amplified by PCR from the PET-HuR plasmid containing full-length HuR. PCR products were digested with restriction endonucleases BamH I and Sal I, and then ligated into pET28a-SUMO plasmid carrying the Ulp1 cleavage site. Recombinant plasmids were confirmed by DNA sequencing and transformed into E. coli BL21 (DE3) to express recombinant proteins with N-terminal 6× His-SUMO tag. E. coli cells were cultured in Luria Bertani (LB) medium at 37 °C with kanamycin (50 mg/L) until the $OD_{600}$ reached 0.6–0.8, then induced with 0.2 mM isopropyl-β-d-thiogalactoside (IPTG) at 18 °C. After 16 h of expression, cells were collected by centrifugation at 6000 r.p.m. for 15 min and further resuspended in buffer containing 20 mM Tris-HCl, pH 8.0, 500 mM NaCl, 20 mM Imidazole, pH 8.0, then lysed using a high-pressure cell disrupter (JNBIO). Cell lysates were centrifuged at 17,000 r.p.m. for 1 h at 4 °C. Supernatants were purified with Ni-NTA (GE Healthcare), then washed with lysis buffer and eluted with buffer containing 20 mM Tris-HCl, pH 8.0, 500 mM NaCl, and 500 mM imidazole. The fusion protein was dialyzed with lysis buffer for 4 h and Ulp1 protease was added to remove the N-terminal 6xHis-SUMO tag. The mixture was applied to another Ni-NTA resin to remove the Ulp1 protease and uncleaved proteins. Eluted proteins were concentrated by centrifugal ultrafiltration, loaded onto a HiLoad 16/60 Superdex 75 column (GE Healthcare) pre-equilibrated with buffer containing 10 mM Tris-HCl, pH 8.0, 100 mM NaCl. Fractions containing purified protein were pooled together and concentrated by centrifugal ultrafiltration. Protein concentrations were determined by UV spectrophotometry with predicted extinction coefficients using Vector NTI.

The RNA oligonucleotides were ordered from Dharmacon (GE Healthcare Dharmacon Inc.) and dissolved in 10 mM Tris, pH 8.0, 100 mM NaCl. ITC analysis of HuR bound to RNA substrates was carried out using a MicroCal iTC200 calorimeter (Marvern Inc.) at 25 °C. The buffer used for proteins and RNA oligomers was 10 mM Tris, pH 8.0, 100 mM NaCl. The concentrations of proteins and RNA were determined by a UV spectrometer. The ITC experiments involved 20–30 injections of protein into RNA substrates. The sample cell was loaded with 250 μl of RNAs at 3 μM and the syringe with 80 μL of protein at 75 μM, in some cases for much better binding curve, the RNA oligomers were diluted to 5 μM and the protein was 100 μM. Reference measurements were carried out to compensate for the heat of dilution of the proteins. Curve fitting to a single-binding site model was performed by the ITC data analysis module of Origin 7.0 (MicroCal) provided by the manufacturer, the titration data were deconvoluted based on a binding model containing CCTA were annealed using a nonlinear least-squares algorithm. Δg° of protein-RNA binding was computed as RTln (1/KD), where R, T and $K_D$ are the gas constant, temperature and dissociation constant, respectively.

**Telomerase activity and telomere length measurement.** Telomere repeat amplification protocol (TRAP) was used for testing telomerase activity[38]. Cells were collected and used for the analysis of telomerase activity. The products were resolved on polyacrylamide gels (8%) and visualized by staining with SYBR Safe (Invitrogen). For telomere length measurement, we used a kit from Roche (Cat. No. 12 209 136 001) and performed southern blot analysis following the manufacturer's instructions. Isolated genomic DNA was digested with Hinf I and Rsa I and resolved on 0.8% agarose gels. The denatured and dried gel were hybridized with Digoxin-labeled, telomere-specific hybridization probe (TAGGG) and exposed to X-ray film. The mean terminal restriction fragment (TRF) length was calculated according to the formula: mean $TRF = \Sigma(OD_i)/ \Sigma(OD_i/L_i)$, where $OD_i$ was the signal intensity and $L_i$ was the molecular weight at the position $i$. All of the uncropped images were included in Supplementary Information.

**Bisulfite RNA sequencing.** For bisulfite RNA sequencing, 1 μg of cellular RNA was dissolved in 10 μl of RNase-free water and mixed with 42.5 μl 5 M sodium bisulfate (Epitect) and 17.5 μl DNA protection buffer (Epitect), incubated in 70 °C for 5 min and 60 °C for 60 min, repeating for 3–5 cycles. Samples were desalted by using Micro Bio-spin6 columns and then de-sulfonated by 1 M Tris (pH 9.0, 1/1, V/V) at 37 °C for 1 h, followed by ethanol precipitation. The bisulfite-converted RNA (0.2 μg) was reverse-transcribed by RevertAid First Strand cDNA Synthesis Kit (Thermo) using primer GTCGTATCCAGTGCAGGGTCCGAGGTATTCGC ACTGGATACGACTCTTAACCTTCCTACCATTCC and subjected to PCR by using following primer pairs: GCGTCTCAACTGGTGTCGTGGAGTCGGCAAT TCAGTTGAGACGCCTTTTCC and CTAACCCTAACTGAGAAGGGTGTAG

for human TERC C106, GCGTCTCAACTGGTGTCGTGGAGTCGGCAATTCA GTTGAGACGCTTCCTCTTCCT and CCACTGCCATTGTGAAGAGTTGGG for human TERC C323, and GCGTCTCAACTGGTGTCGTGGAGTCGGCAATTCA GTTGAGACGCACTTTCCT and ACCTAACCCTGATTTTTATTAGTTGTGGG TT for mTERC C64. The PCR products were inserted into the pGEM-T Easy Vector System (Promega). The plasmids prepared from single clones were sequenced and aligned with TERC sequence. The cytosine retained was considered to be methylated.

**Mice.** The Terc$^{+/-}$ mice (Ly5.2 strains) were a gift from Dr. K. L. Rudolph. mTERC$^{+/-}$ mice were crossed to produce third-generation G3 mTERC$^{-/-}$ mice. The recipient mice used in the transplantation assays were either CD45.1 mice or CD45.1/45.2 mice. Lethal irradiation dose of the recipient animals was 10 Gy. All mice were maintained in C57BL/6J background. Mice were maintained on a standard diet in 12-h-light and 12-h-dark shifts in Hangzhou Normal University. All of the animal experiments were performed in accordance with the National Institutes of Health Guide for the Care and Use of Laboratory Animals, and with the approval of Animal Care and Use Committee of School of Medicine at Hangzhou Normal University.

**Purification of the bone marrow derived HSCs.** The bone marrow derived HSCs were purified[39–41] are as follows:

1. Mice were killed by cervical dislocation, in accordance with relevant authorities' guidelines and regulations.
2. Using small scissors and forceps, dissected out femurs, tibias, ilium, pelvis, spine, and sternum from C57/B6 mice and placed samples in a 60 mm tissue culture dish containing 4 ml ice-cold PBS.
3. Whole bones can be crushed using a mortar and pestle.
4. Filter cells through 50 mm nylon mesh into a 15 ml tube.
5. Count nucleated cells with a hemacytometer. Usually, $10^8$ bone marrow cells can be obtained from each mouse.
6. Spin down cells for 5 min at 440 g at 4 °C.
7. Aspirate the supernatant, carefully leaving ~30 μl staining medium above the pellet, then resuspend cells in this residuum.
8. Add 5 μl APC-conjugated antibodies against cKit per $10^8$ cells on ice for 30 min.
9. Wash cells with 1 ml PBS by spinning down for 5 min at 440 g at 4 °C, add 5 μl of anti-APC-conjugated magnetic beads (Miltenyi) per $10^8$ cells to stain cells for 15 min at room temperature. Usually, the enrichment of LSK cells using MACS based cKit$^+$ selection is 20-fold, i.e., 2% of LSK cells in the MACS enriched cells.
10. Wash cells with 1 ml PBS by spinning down for 5 min at 440 g at 4 °C, then add 3 μl lineage marker Ab cocktail (CD4, CD8, B220, Ter119, Mac1, Gr1) per $10^7$ cells on ice for 30 min.
11. Wash cells with 1 ml PBS by spinning down for 5 min at 440 g at 4 °C, then stain cells with cKit-APC, Sca1-PE-Cy7, and streptavidin-PE conjugated rat anti-mouse antibodies on ice for 30 min.
12. Wash cells with 1 ml PBS by spinning down for 5 min at 440 g at 4 °C, then sort cells by Lin$^-$c-Kit$^+$Sca1$^+$. BD Influx$^{TM}$ Cell Sorter (http://static.bdbiosciences.com/documents/BD Influx User Guide.pdf) with 86 μm nozzle was used for cell sorting. With the flow rate (drops per second) of 8000 events per second, 5000 LSK cells could be obtained per minute.
13. Sorted LSK cells were cultured in 48-well plate with lentivirus, and were transplanted into lethally irradiated recipients 12 h after infection. The total number of the cells transplanted is about $2 \times 10^4$. After calculation, ~4–6 × $10^3$ GFP$^+$ LSK HSCs were transplanted per mouse in LV-TREC (or variants) expressing group or 1.2–1.8 × $10^4$ GFP$^+$ LSK HSCs were transplanted per mouse in LV-shHuR expressing group.

**Lentivirus production and transduction.** Small hairpin RNA of murine HuR was cloned into the SF-LV-EGFP vector (a kind gift from Lenhard Rudolph lab). Lentivirus was produced in 293T cells after transfection of 14 mg shRNA plasmid, 10 mg pspAX$_2$ helper plasmid and 6 mg pMD2G by using polyethylenimine. Cells were replaced with fresh medium 12 h after transfection. Collected the virus supernatant 48 h after transfection and concentrated by centrifuging the medium at 25,000 r.p.m. for 2.5 h at 4 °C. The virus pellet was dissolved in sterile PBS. Lentiviral titers were assessed and calculated by the transfection of cKit positive cells. A total of $1 \times 10^5$ sorted LSK cells were transduced with the Lentivirus[42]. The transduction efficiency was 20–30% in LV-TREC (or variants) expressing group or 60–90% in LV-shHuR expressing group (Supplementary Fig. 11).

**Flow cytometry.** Antibodies used for flow cytometry analysis or cell sorting were purchased from eBioscience, Biolegend, or BD Bioscience. Total bone marrow cells were stained with surface markers and sorted by flow cytometry. To quantify dead cells, DAPI (LifeTechnologies) was added before flow cytometry. Samples were analyzed on LRSFortessa (BD Biosciences) or sorted on Influx (BD Biosciences). The gating strategy for FACS analysis and sorting is shown in Supplementary Figs. 12, 13.

**Real-time qPCR assays for telomere measurement**. One hundred cells were sorted into lysis buffer (1/200 Proteinase K in T10E1 buffer). The sorted cells were lysed using a heat block: 1 h at 50 °C and 4 min at 99 °C. The reaction was set up with SYBR Green I in 96-well plates. Run primers against the telomeric repeats and a reference single-copy genomic gene *36B4* to allow direct input normalization across samples. Telomere forward: CGGTTTGTTTGGGTTTGGGTTTGGGTTTG GGTTTGGGTT; telomere reverse: GGCTTGCCTTACCCTTACCCTTACCCTTA CCCTTACCCT (36B4 forward: AGATTCGGGATATGCTGTTGGC/36B4 reverse: TCGGGTCCTAGACCAGTGTTC). The relative telomerase length was calculated as a ratio of telomere repeats to single-copy gene (SCG) copies (T/S ratio).

**Double-stranded RNA pull-down assays**. Double-stranded RNA (dsRNA) pull-down assays were performed as reported[43]. Briefly, HeLa cells were transfected with a siHuR or a control siRNA. Forty-eight hours later, cells were collected by incubating in J2 lysis buffer (25 mM HEPES (pH 7.2), 75 mM NaCl, 5 mM MgCl$_2$, 0.5% NP40, 1 U RNasin) on ice for 30 min. The samples were sonicated (5 min, 30 s ON, 30 s OFF) and centrifuged at 12,000 r.p.m. for 20 min at 4 °C. Protein G Dynabeads (25 μl, pre-washed twice with J2 lysis buffer) and J2 mAb (2 μg, Sci-cons) or mouse IgG (2 μg, Santa Cruz) were incubated in 350 μl J2 lysis buffer for 1 h at 4 °C, washed twice with J2 lysis buffer and then were added to the sample volume of clarified lysates (300 μg), followed by rotation for 3 h at 4 °C. The beads were then washed three times for 5 min at 4 °C with 350 μl of J2 lysis buffer. RNA isolated from the pull-down materials was subjected to RT-qPCR analysis to assess *TERC* dsRNA. A total of 10% of input lysates were used as a reference for each of the pull-down assays.

**Data availability**. The data that support the findings of this study are available from the corresponding authors upon reasonable request.

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

# Acknowledgements

This work was supported by Grant 2017YFA0504300 and 2017YFA0103302 from the National Key Research and Development Program of China; Grants, 81420108016, 91749208, 91749203, 81525010, 81420108017, 81372166, and 81572843 from the National Natural Science Foundation of China. M.G. was supported by the NIA IRP, NIH. We thank Dr. K. L. Rudolph for providing *Terc ±* mice.

# Author contributions

W.W., Z.J., J.M., Y.D., W.M., Y.Z., X.W., Z.S. and M.G. participated in designing the study. H.T., X.C., X.F., W.M., Y.C., Y.T., M.Z., C.L., D.S., Y.C., J.M., J.X., X.Z., X.Y. and B.

J. performed the in vitro and cell culture experiments. H.W., F.Y. and Z.J. performed the stem cell experiments. W.W., Z.J. and M.G. wrote the paper. Y.C. provided the p3× Flag-CMV 10 vector expressing flag-hTERT (which expressed full-length human *TERT* mRNA), M.G. provided the pSL-MS2 vector, and W.M. provided the FG-EH-DEST2-YFPc-ppw vector.

## Additional information

**Competing interests:** The authors declare no competing interests.

