## [Peer Review File · Nature Communications]

Reviewers' comments:

Reviewer #1 (Remarks to the Author):

This is a manuscript I find hard to review. The authors have shown binding of HuR to TERC and effects of HuR to telomere synthesis but did not really provide a mechanism. They seem to suggest HuR facilitates m5C methylation on C106, but as far as I can tell it is unclear how C106 methylation (at 30%) could affect telomere synthesis. The authors also do not know how HuR facilitates C106 methylation. There are several gaps in between. The authors will need to identify methyltransferase can connect HuR to C106 methylation and KD of this methyltransferase to show expected effect. Or figure out how C106 methylation facilitates telomere synthesis.

Reviewer #2 (Remarks to the Author):

Tang et al. report that HuR regulates telomerase activity through methylation of TERC. They show that exogenous expression of HuR interacts with in vitro transcribed TERC, and that specific mutations and methylation of TERC at specific residues affects this interaction. They test whether exogenous tagged TERC interacts with HuR in cells using TriFC, IF/FISH and IP. They also report that HuR – similar to a positive control protein, hnRNPC - interacts with active telomerase, while a negative control protein, p53, does not. They show that shRNA against HuR leads to a loss of telomerase activity and telomere length - presumably due to the loss of TERC association/expression with TERT that they show. They show that specific mutations in TERC, as well as loss of HuR expression, leads to reduced methylation of TERC at C106. They report that in mTERC^{+/-} mice, shRNA against HuR leads to telomere loss and reduced HSC maintenance.

This study addresses an interesting and important topic in telomere biology and I believe the scope of the study could potentially be of interest to readers in that field and in particular those studying Dyskeratosis congenita, and other bone marrow failure and premature aging syndromes. The paper does not reveal a significant conceptual advance regarding the biology telomerase RNA, but represents important observations about a novel interaction partner with potentially significant functional implications for the biology of telomerase and disease. For example, anything that may improve HuR binding to TERC may improve the outcome of patients suffering from pertinent TERC mutations. Alternatively, if cancer cells rely on telomerase, specific inhibition of TERC with HuR may inhibit telomere maintenance in cancer cells.

Overall, I found that the figure legends would benefit from a much more thorough description of the data. There are a number of important controls that I think are essential to add to the study for the authors to safely make their conclusions.

1. Fig 1 D requires a non-binding negative RNA control substrate.
2. For Fig 2B FACS I would like to see a superior negative control included in the study. One example would be a nuclear protein, rather than a cytosolic protein that is membrane embedded. The current negative control provides minimal confidence that the interaction between HuR and TERC is specific within the nucleus; specifically, a nuclear RNA binding protein that does not bind TERC would be ideal.
3. Fig 2C, the HuR staining and TERC FISH data are not convincing at all due to poor signal above

background, and needs to be improved before publishing. Also, there is a typo – “Collin” should read “coilin”.

4. What hnRNPC isoform and what antibody were detected/used? Why is so much of the telomerase associated with hnRNPC? It should be a small fraction based on studies from the Collins lab. This leads me to question the quantification using TRAP assay; typically, the TRAP assay is not considered to be a highly quantitative assay for telomerase. Reliable quantitative assessments of the impact of mutations on telomerase activity really should be done using a "direct telomerase assay" (Cohen & Reddel 2008 Nat Methods) – at least for some of the most critical findings.

5. For the experiments done in cells, I would like to see included in the study Northern blots or RT-PCR for TERC levels after HuR shRNA or expression of TERC mutations vs wt.

6. Using qPCR to measure telomere length is not standard and I would like to see it compared to a "TRF southern blot" (published or otherwise) for at least two cell samples that vary in telomere length. The authors already have these DNA samples so this should be relatively easy for them to perform this control.

Reviewer #3 (Remarks to the Author):

The manuscript by tang et al is a through evaluation of the role of the RNA binding protein HuR on telomerase activity regulation. They report that HuR regulates telomerase activity by methylation of TERC. The data is convincing in some places, however quite dubious in others. The following are a list of my issues.

The authors are studying the ELAV RNA binding protein HuR-- What does 'HuR' stand for..? This needs to be spelt out early on in the manuscript.

What does ELAV stand for in the Intro ?

Figure 2—What is PE and FITC signaling in panel B ?

C) State somewhere that Coilin is a Cajal body marker...

Major issue-- Data not convincing, particularly the TERC-HuR association; need to show more cells with co-localization, not just one

Figure 3B- Was the same extract used for all IP assays ?

Figure 4B- Y-axis should be telomere length, not telomerase

4C, 4D—How was TERC levels measured ? Need to state in Figure legend.

Figure 5. TRAP assay results for panels C and D should be combined into 1 panel

Figure 7— Major issues throughout.

The transplantation experiment is confusing. How many LSK HSC were transduced with the LV vector, and what is the transduction efficiency (% GFP +ve cells just prior to transplant) ?

Where were the Terc +/- mice obtained from ? Most strains are on a LY5.1 background, not Ly5.2...

How many GFP+ LSK HSC were transplanted per mouse ?

The conditions for lethal irradiation of the recipient animals needs to be provided.

Major- HSC are notoriously difficult to transduce; the cytometry data for LV transduction efficiency just prior to transplant must be shown, as well as the FACS data for sorting the LT-HSC (LSK HSC/Fit3+/CD34-).

In which cells was TL analysis done in ? LSK-HSC ? WBM ?

Minor issues- Add space between 'The' and 'TRAP' line 144.

A general grammar check needs to be done throughout the manuscript. There are many grammatical errors throughout.

Response to the comments of reviewer 1:

This is a manuscript I find hard to review. The authors have shown binding of HuR to TERC and effects of HuR to telomere synthesis but did not really provide a mechanism. They seem to suggest HuR facilitates m5C methylation on C106, but as far as I can tell it is unclear how C106 methylation (at 30%) could affect telomere synthesis. The authors also do not know how HuR facilitates C106 methylation. There are several gaps in between. The authors will need to identify methyltransferase can connect HuR to C106 methylatoin and KD of this methytransferase to show expected effect. Or figure out how C106 methylation facilitates telomere synthesis”.

We appreciate the reviewer's comments.

There are four concerns raised from reviewer 1.

1. how C106 methylation (at 30%) could affect telomere synthesis?

The formation/assembly of holoenzyme of telomerase is an essential step for telomere synthesis. We found that the methylation of *TERC* facilitated the assembly of TERT/*TERC* complex, thereby affecting the telomerase activity and telomere length. Moreover, the methylation of TERT-bound *TERC* is much higher than that of unbound *TERC* (Fig. 5e). It should be noted that due to the low efficiency of IP assays, the actual methylation rate of TERT-bound *TERC* is likely to be much higher than that measured. Together, our results suggest that methylated *TERC* affects telomere synthesis by facilitating the assembly of TERT/*TERC* complex.

2. The authors also do not know how HuR facilitates C106 methylation.

Thus far, the mechanisms underlying the effect of RNA-binding proteins (RBPs) on the fate of the target RNA are largely unknown. RBPs can recruit or compete with other proteins or nucleic acids to modulate their interaction with the target RNA, and/or they can elicit local changes in the conformation of the bound RNA. Thus, HuR may facilitate the assembly of TERT/*TERC* complex formation or *TERC* methylation by influencing the conformation of *TERC*. However, obtaining evidence to support changes in the structure of this complex would require advanced structural methodologies such as crystallography or cryo-EM which are outside of the realm of this study.

3. The authors will need to identify methyltransferase can connect HuR to C106 methylatoin and KD of this methytransferase to show expected effect.

[Redacted]

4. or figure out how C106 methylation facilitates telomere synthesis.

This concern has been addressed in the response to concern 1.

Response to the comments of reviewer 2:

Tang et al. report that HuR regulates telomerase activity through methylation of TERC. They show that exogenous expression of HuR interacts with in vitro transcribed TERC, and that specific mutations and methylation of TERC at specific residues affects this interaction. They test whether exogenous tagged TERC interacts with HuR in cells using TriFC, IF/FISH and IP. They also report that HuR – similar to a positive control protein, hnRNPC - interacts with active telomerase, while a negative control protein, p53, does not. They show that shRNA against HuR leads to a loss of telomerase activity and telomere length - presumably due to the loss of TERC association/expression with TERT that they show. They show that specific mutations in TERC, as well as loss of HuR expression, leads to reduced methylation of TERC at C106. They report that in mTERC^{+/-} mice, shRNA against HuR leads to telomere loss and reduced HSC maintenance.

This study addresses an interesting and important topic in telomere biology and I believe the scope of the study could potentially be of interest to readers in that field and in particular those studying Dyskeratosis congenita, and other bone marrow failure and premature aging syndromes. The paper does not reveal a significant conceptual advance regarding the biology telomerase RNA, but represents important observations about a novel interaction partner with potentially significant functional implications for the biology of telomerase and disease. For example, anything that may improves HuR binding to TERC may improve the outcome of patients suffering from pertinent TERC mutations. Alternatively, if cancer cells rely on telomerase, specific inhibition of TERC with HuR may inhibit telomere maintenance in cancer cells.

Overall, I found that the figure legends would benefit from a much more thorough description of the data. There are a number of important controls that I think are essential to add to the study for the authors to safely make their conclusions.

We thank reviewer 2 for his/her positive comments.

1. Fig 1 D requires a non-binding negative RNA control substrate

The reviewer is correct that we needed to include a negative control. As shown in the revised manuscript (Supplementary Fig. S2), RNA fragment CUAGACUAGGACUC, which does not bind with HuR, was used as a negative control for ITC assays.

2. For Fig 2B FACS I would like to see a superior negative control included in the study. One example would be a nuclear protein, rather than a cytosolic protein that is membrane embedded. The current negative control provides minimal confidence that the interaction

between HuR and TERC is specific within the nucleus; specifically, a nuclear RNA binding protein that does not bind TERC would be ideal.

We agree with the reviewer's suggestion. In the revised manuscript, we included the analysis of an RNA-binding protein that localizes in the nucleus (PABPC1) as a negative control. These data were added to the revised Fig. 2b.

3. Fig 2C, the HuR staining and TERC FISH data are not convincing at all due to poor signal above background, and needs to be improved before publishing. Also, there is a typo – "Collin" should read "coilin".

In the revised manuscript, we replaced the HuR staining and TERC FISH data with more clear images (3 images for each). Because the figures were too large, we moved them to supplementary information (Supplementary Fig. S3). The spelling error was corrected.

4. What hnRNPC isoform and what antibody were detected/used? Why is so much of the telomerase associated with hnRNPC? It should be a small fraction based on studies from the Collins lab. This leads me to question the quantification using TRAP assay; typically, the TRAP assay is not considered to be a highly quantitative assay for telomerase. Reliable quantitative assessments of the impact of mutations on telomerase activity really should be done using a direct telomerase assay"(Cohen & Reddel 2008 Nat Methods) – at least for some of the most critical findings.

Figure 3 for reviewer 2. TRAP assays using IP materials (9 independent experiments).

We appreciate these questions. The antibody used (from Proteintech Corp., catalog number 11760-1-AP) recognized both hnRNPC1 and hnRNPC2. The telomerase activity detected in the IP materials is similar to that previously shown by Jerry Shay's lab (Ford et al, 2000, Mol. Cell. Biol., 20: 9084-9091); in these earlier studies, the telomerase activity observed from hnRNPC IP materials was also quite strong. This TRAP assay was repeated numerous times in the present study. As shown in Figure 3 for reviewer 2, the high telomerase activity present in the hnRNPC IP materials is quite reproducible.

Figure 4 for reviewer 2. Telomerase assays following the method reported by Gregg B. Morin (Gregg B. Morin. The human telomere terminal transferase enzyme is a ribonucleoprotein that synthesizes TTAGGG repeats. Cell. 59, 521-529, 1989). Biotin-11-dGTP (PerkinElmer, NEL541), biotinylated telomere primer 5'-biotin-CTAGACCTGTCATCA(TTAGGG)₃-3' were used in the telomere repeat synthesis reactions. Lysates were prepared following methods reported by Sun et al (Cancer Research Techniques, 25, 1046-51, 1998). The products were analyzed on 8 M urea-8% polyacrylamide gels (length, 8 cm), transferred to NC membrane (following instruction of Light Shift Chemiluminescent EMSA Kit, Thermo Scientific, 20418). The membrane was incubated with an anti-biotin antibody and the signal was further visualize same as the procedures for Western blot analysis.

We purchased ³²P-labeled dGTP from Perkin Elmer to perform the telomerase assay, but failed to get any signal, likely because of the slow transit through customs and the short half-life of ³²P (this isotope was not available for purchase in China during the review period). Therefore, we opted instead for performing “direct telomerase assay” by using biotinylated dGTP and biotinylated telomere primer 5'-biotin-CTAGACCTGTCATCA (TTAGGG)₃-3' following the procedures reported by Gregg B. Morin. As shown in ‘Figure 4 for reviewer 2’, the telomerase activity obtained from hnRNPC IP materials was pretty strong, in agreement with the findings from TRAP assays (Fig. 3b). In addition, *TERC* variants bearing U40A+U100A or C106G gave lower telomerase activity, as shown by using TRAP assays (Fig. 5c). We tried this experiment numerous times but only observed short repeats, in agreement with the strong signals reported previously in short repeats but not in longer ones (Gregg B. Morin; Scott B. Cohen and Roger R. Reddel). Despite many optimization efforts, we were unable to observe the longer repeats reported by the Morin lab (Cell. 59, 521-529, 1989), even though the results support the results from the TRAP assays shown in our manuscript. Thus, we would prefer to show these results only for the reviewer’s benefit (‘Figure 4 for reviewer 2’).

5. For the experiments done in cells, I would like to see included in the study Northern blots or RT-PCR for TERC levels after HuRshRNA or expression of TERC mutations vs wt.

We thank the reviewer for this important request. In the revised Fig. 4c, *TERC* levels are shown in cells expressing HuR shRNA. We also included RT-PCR analysis for the levels of *TERC* and variants bearing U40A, U100A, U40A+U100A, C106G, G107U, GC107/108AG,

C166U, or C445U in TRAP assays (revised Supplementary Fig. S5c, Fig. S7, Fig. S8b, and Fig. S9c).

6. Using qPCR to measure telomere length is not standard and I would like to see it compared to a "TRF southern blot" (published or otherwise) for at least two cell samples that vary in telomere length. The authors already have these DNA samples so this should be relatively easy for them to perform this control.

Because HSC numbers are very low, we use qPCR analysis to measure telomere length following the method described in Beerman et al., Cell Stem Cell (2014). As requested by the reviewer, we measured telomere length of the WT and G3Terc^{-/-} MEFs using "Flow-FISH". The data are shown below (Figure 5 for reviewer 2). If the reviewer prefers that we include them in the manuscript, we will be happy to do that.

Figure 5 for reviewer 2. Telomere length was measured by Flow-FISH in WT and G3Terc^{-/-} MEF cells. The telomere probe signals of WT MEF cells were significantly higher than those of G3Terc^{-/-} MEF cells. Methods: Sorted myeloid cells were hybridized and stained using the DAKO Telomere PNA kit (Denmark). Briefly, on a single cell suspension, the sample DNA was denatured at 82°C for 10 min in a microcentrifuge tube either in the presence of hybridization solution without a probe, or in hybridization solution containing a Cy3-conjugated PNA telomere probe. Hybridization then took place in the dark at room temperature overnight. The hybridization was followed by two 10 min post-hybridization washes at 40°C. Flow cytometry measurement was performed by FACS Fortessa.

A previous study comparing the qPCR with "TRF Southern blot" methods (Yang Z, Huang X, Jiang H, Zhang Y, Liu H, Qin C, Eisner GM, Jose PA, Rudolph KL, Ju Z. Short Telomeres and Prognosis of Hypertension in a Chinese Population. Hypertension, 53, 639-45, 2009) deemed qPCR analysis adequate to measure telomere length in HSCs.

Response to the concerns from reviewer 3:

The manuscript by tang et al is a through evaluation of the role of the RNA binding protein HuR on telomerase activity regulation. They report that HuR regulates telomerase activity by methylation of TERC. The data is convincing in some places, however quite dubious in others. The following are a list of my issues.

We appreciate the positive comments from reviewer 3 and his/her insightful comments.

The authors are studying the ELAV RNA binding protein HuR-- What does 'HuR' stand for..? This needs to be spelt out early on in the manuscript. What does ELAV stand for in the Intro ?

We apologize for this oversight. In the revised manuscript, we have defined ELAV and HuR in the Introduction section. ELAV stands for 'embryonic lethal abnormal vision' in *Drosophila*, and HuR (or ELAV-like 1, ELAVL1) stands for 'human antigen R'.

Figure 2—What is PE and FITC signaling in panel B ?

We appreciate the reviewer's pointing out these errors. The Y-axis should be FL2 (not PE) and the X-axis should be FL1-YFP (not FITC). We have corrected them in the revised Fig. 2B.

C) State somewhere that Coilin is a Cajal body marker...

We appreciate these requests. In the revised manuscript, we state that Coilin is a Cajal body marker. "Consistent with this finding, we observed the co-localization of HuR and *TERC* (Supplementary Fig. S3a), as well as the localization of HuR in Cajal bodies, as evidenced by the co-localization of HuR and Coilin (a marker of Cajal bodies; Supplementary Fig. S3b), where *TERC* undergoes maturation".

Major issue-- Data not convincing, particularly the TERC-HuR association; need to show more cells with co-localization, not just one

In response to this important request, also made by reviewer 2, we now show 3 images for each co-localization assays.

Figure 3B- Was the same extract used for all IP assays?

Yes, we used same extracts for all IP assays..

Figure 4B- Y-axis should be telomere length, not telomerase

Thank you for catching this error, we have corrected it.

4C, 4D—How was TERC levels measured ? Need to state in Figure legend.

We have described this in the figure legend.

Figure 5. TRAP assay results for panels C and D should be combined into 1 panel

In the revised manuscript, we have redone the TRAP assays and have combined Fig. 5C and

D.

Figure 7— Major issues throughout. The transplantation experiment is confusing. How many LSK HSC were transduced with the LV vector, and what is the transduction efficiency (% GFP +ve cells just prior to transplant) ?

1×10^5 purified (sorted from mouse bone marrow) LSK cells were transduced with the concentrated lentivirus, it is indicated in the “Methods” section. The FACS analysis monitoring the transduction efficiency in LSK cells expressing HuR shRNA (60-90%) and mTERC or its variants (20-30%) was shown in an additional Supplementary Figure (Supplementary Fig. S10). The efficiency of the purification (sorting) of LSK cells from mouse bone marrow was also analyzed by FACS and shown in Supplementary Fig. S11.

Where were the Terc +/- mice obtained from? Most strains are on a LY5.1 background, not Ly5.2...

The Terc^{+/-} mice were Ly5.2 strains from KL Rudolph’s lab; this has been indicated in the “Methods” section.

How many GFP+ LSK HSC were transplanted per mouse ?

The total number of the cells transplanted is about 2×10^4 . After calculation, approximately $4-6 \times 10^3$ GFP+ LSK HSCs were transplanted per mouse in LV-TREC (or variants) expressing group or $1.2-1.8 \times 10^4$ GFP+ LSK HSCs were transplanted per mouse in LV-shHuR expressing group. This information has been included in the revised “Methods” section.

The conditions for lethal irradiation of the recipient animals needs to be provided.

The lethal irradiation dose of the recipient animals was 10 Gy. This dose is indicated in the revised “Methods” section.

Major- HSC are notoriously difficult to transduce; the cytometry data for LV transduction efficiency just prior to transplant must be shown, as well as the FACS data for sorting the LT-HSC (LSK HSC/Flt3+/CD34-).

It is true that HSC are difficult to transduce, therefore we have transduced the LSK HSCs with highly concentrated lentivirus (titer $>10^8$ pfu/ml). In the revised manuscript, we have included the transduction efficiency and the FACS data for sorting the LSK HSC from bone marrow in the additional Supplementary Fig. S10-S11.

In which cells was TL analysis done in ? LSK-HSC ?WBM ?

The Telomere Length analysis was done in LT-HSCs of donor, as described in the revised Figure legend for Figure 7.

Minor issues- Add space between 'The' and 'TRAP' line 144.

A general grammar check needs to be done throughout the manuscript. There are many grammatical errors throughout.

We have corrected the grammatical errors throughout the manuscript.

Reviewers' comments:

Reviewer #1 (Remarks to the Author):

The authors have made efforts to address comments from reviewers. I appreciate the challenge to identify methyltransferase. I would hope to see how HuR facilitates C106 methylation and how the methylation promotes assembly of TERT/TERC complex. At least some mechanistic studies or hypothesis.

Reviewer #2 (Remarks to the Author):

I thank the authors for addressing the concerns raised in the initial review and I like some of the new data. However, I have additional concerns about other new data and I cannot recommend this for publication without addressing these issues.

1. Can the authors please provide a full description of the RT-PCR protocol, including all reagents, the PCR program, normalizations, and qPCR machine used for the assessment of TERC levels. This information is critical for interpreting the results.

2. Also with regard to the RT-PCR, the authors state that the oligo used for reverse transcription was:

GCGTCTCAACTGGTGTCTGGAGTCGGCAATTCAGTTGAGACGCGCATGTGTGAG

This appears to be a gene-specific RT primer ? Although a BLAST search does not identify any sequence associated with this oligo. Can the authors please indicate why those chose this sequence for the RT step.

qRT-PCR should be normalized, typically to a transcript from a 'house keeping' gene such as GAPDH. The primers generally used for reverse transcription are random hexamers or oligo dT (or both). Thus, it is clear that authors have not normalized the TERC transcripts to such a control. Could the authors please explain why this was not done?

3. Possibly more importantly, the primer above used for the RT step is not present in the 451 nucleotide human TERC transcript which is:

GGGTTGCGGAGGGTGGGCCTGGGAGGGGTGGTGGCCATTTTTGTCTAACCTAACTGAGAAGGG
CGTAGGCGCCGTGCTTTTGTCTCCCGCGCGCTGTTTTCTCGCTGACTTTCAGCGGGCGGAAAAG
CCTCGGCCTGCCGCTTCCACCGTTCATTCTAGAGCAAACAAAAATGTCAGCTGCTGGCCCGTT
CGCCCTCCCGGGGACCTGCGGCGGGTGCCTGCCAGCCCCGAACCCCGCCTGGAGGCCGCGG
TCGGCCCGGGGCTTCTCCGGAGGCACCCACTGCCACCGCGAAGAGTTGGGCTCTGTAGCCGCGG
GTCTCTCGGGGGCGAGGGCGAGGTTTCAGGCCTTTCAGGCCGAGGAAGAGGAACGGAGCGAGTCC
CCGCGCGCGGCGGATTCCCTGAGCTGTGGGACGTGCACCCAGGACTCGGCTCACACATGC

Therefore, it is unclear how the PCR was successful if the incorrect RT primer was used. Could the authors please explain this?

One possibility is that the authors did not use DNase treatment, and subsequently did not include a 'no-reverse transcriptase' control in the RT reactions to assess DNA contamination in the PCR. If so, this would account for all of the fairly uniform qPCR results.

4. While the forward primer used for TERC qPCR (TTCAGGCCTTTCAGGCCGAGGAA) is present in

this TERC sequence (underlined above) the reverse complement of the reverse primer used (TGGTGTTCGTGGAGTCGGC) is not present in this sequence. Thus, it is not clear how the PCR worked for TERC? Did the authors sequence verify the PCR product??

As far as can tell, collectively these points appear to invalidate the qPCR results. Thus, an explanation addressing each of these would be necessary for me to recommend this manuscript to be taken further.

5. Could the authors please perform the TriFC experiment with the TERC C106 mutant? This would provide a conclusive result that the observed HuR interaction is mediated by C106 - as well as more fully validate the TriFC method to address this question.

6. Is the legend for Fig S4C correct?

Reviewer #3 (Remarks to the Author):

As pointed out by other reviewers, this is a thorough evaluation of the effect of RNA binding protein HuR on regulation of telomerase.

I have one major concern, regarding the authors response to Reviewer #3 on the number of GFP+ LSK HSC transplanted. The numbers of transplanted cells stated in the response, namely $4-6 \times 10^3$ or $1.2 - 1.8 \times 10^4$ seem very unrealistic to me, as this would correspond to the ENTIRE HSC population from the 4 long bones of at least 2-3 adult mice. More concerning is the purported number of LSK sorted, 1×10^5 , which correspond to about 20-25 mice worth of HSC. This would be an extremely time consuming and costly endeavor. Furthermore it would literally take days to FACS sort this many LSK HSC.

How many mice were used as LSK HSC donors in this experiment?

How long did it take to sort the LSK HSC ?

The FACS data for sorting the LSK HSC must be shown, at least as a supplemental figure.

I cannot support publication of this paper until these 2 questions are answered, and I have seen the FACS data (dot plot).

Response to the Reviewers' comments:

Reviewer #1 (Remarks to the Author):

The authors have made efforts to address comments from reviewers. I appreciate the challenge to identify methyltransferase. I would hope to see how HuR facilitates C106 methylation and how the methylation promotes assembly of TERT/TERC complex. At least some mechanistic studies or hypothesis.

We agree with the reviewer that understanding how HuR facilitates C106 methylation and how such methylation promotes assembly of TERT/TERC complex would greatly improve our manuscript. As indicated at <http://telomerase.asu.edu/diseases.html>, the secondary structure of *TERC*, especially in its pseudoknot/template domain, is of critical importance for the assembly of TERT/TERC complex. By using a recently reported method [Aktaş et al. 'DHX9 suppresses RNA processing defects originating from the Alu invasion of the human genome'. *Nature* **544**, 115-119 (2017)], we tested if the association of HuR with *TERC* influenced the secondary structure [i.e., the formation of double-stranded (ds)RNA] of *TERC*; we have included the data in the revised Supplementary Information. As shown in Fig. S14, knockdown of HuR reduced the levels of *TERC* dsRNA. Therefore, we hypothesized that HuR might promote the assembly of the TERT/TERC complex by influencing the formation of *TERC* dsRNA segments. Once the methyltransferase responsible for methylating *TERC* is identified, we will be able to measure if methylation of *TERC* could exert similar effect by using same method.

In the revised Discussion section, we have included this information, along with the hypothesis, as requested: "The finding that knockdown of HuR reduces the formation of double-stranded *TERC* RNA (Fig. S14) suggests that the association of HuR with *TERC* may enhance *TERC* C106 methylation by altering the secondary structure of *TERC*; the influence of *TERC* methylation on the secondary structure of *TERC* remains to be further studied".

Reviewer #2 (Remarks to the Author):

I thank the authors for addressing the concerns raised in the initial review and I like some of the new data. However, I have additional concerns about other new data and I cannot recommend this for publication without addressing these issues.

1. *Can the authors please provide a full description of the RT-PCR protocol, including all reagents, the PCR program, normalizations, and qPCR machine used for the assessment of TERC levels. This information is critical for interpreting the results.*

We appreciate this request. To measure the levels of *TERC*, we first treated the samples with DNase I and then used a reverse primer for the reverse transcription reaction that contains sequences that do not exist in the 451 nt of *TERC* sequence, ensuring that the PCR products were amplified from *TERC* RNA. The levels of *U6* were measured in order to normalize the results from all of the reverse transcription and real-time qPCR analyses. Since *U6* is abundant, it served as a normalization marker in pulldown experiments that required measurement of nonspecific binding.

As requested by the reviewer, details of the protocol for RT-PCR analysis are provided (included in the Supplementary Information):

1. Add into sterile, nuclease-free tube on ice in the indicated order (12 μ l in total): nuclease-free water, DNase-treated total RNA (1 μ g), stem-loop gene-specific RT primer (20 pmol).
2. Mix gently, centrifuge briefly and incubate with the following program in a thermal cycler: 65 $^{\circ}$ C (5 min), 45 $^{\circ}$ C (5 min), 25 $^{\circ}$ C (5 min), 4 $^{\circ}$ C (hold).
3. Add the following components in the indicated order: 5 X Reaction Buffer (4 μ l), RNase inhibitor (1 μ l; 20 U, Thermo Scientific, EO0381), dNTP Mix, 10 mM each: 2 μ l (1 mM final concentration, Thermo Scientific, R0191), reverse transcriptase: 1 μ l (200 U, Thermo Scientific, EP0441). Mix gently and centrifuge briefly.
4. Using a thermal cycler, proceed reverse transcription with the following program: 25 $^{\circ}$ C (10 min), 42 $^{\circ}$ C (60 min), 70 $^{\circ}$ C (10 min), 4 $^{\circ}$ C (hold).
5. Perform qPCR assays in the following programs (Biorad CFX96 real-time system): pre-denature at 95 $^{\circ}$ C for 10 min, use [95 $^{\circ}$ C (15 s), 60 $^{\circ}$ C (45 s)] for 40 cycles, then perform melting curve analysis and normalize using comparative Ct ($\Delta\Delta$ Ct) quantification method.

2. Also with regard to the RT-PCR, the authors state that the oligo used for reverse transcription was:

CGGTCTCAACTTGGTGTCTGGAGTCGGCAATTCAGTTGAGACGC**GCATGTGTGAG**

This appears to be a gene-specific RT primer? Although a BLAST search does not identify any sequence associated with this oligo. Can the authors please indicate why those chose this sequence for the RT step.

The reviewer is correct, we employed a stem-loop gene-specific primer for RT. Given that the sequence of mature *TERC* transcript is identical to that of its genomic DNA counterpart, the cDNA produced from traditional RT process with oligo dT and/or random hexamers may not be distinguishing from genome DNA. The stem-loop RT primer has been extensively used for the qPCR analysis of long non-coding RNAs and microRNA (see for example, Salone V and Rederstorff M. Stem-loop RT-PCR based quantification of small non-coding RNAs. *Methods Mol Biol*. 2015. 1296:103-8; and Tong et al., Improved RT-PCR Assay to Quantitate the Pri-, Pre-, and Mature microRNAs with Higher Efficiency and Accuracy. *Mol Biotechnol*. 2015 57(10):939-46)). The stem-loop RT primer was capable of distinguishing microRNA from its primary microRNA or precursor microRNA sequences.

The sequences matching the 3' end of *TERC* and the reverse primer of real-time qPCR are indicated in red and underlined, respectively. We also indicated that the reverse primer sequence matched the sequence of the primers for reverse transcription in the "Methods" section.

qRT-PCR should be normalized, typically to a transcript from a 'housekeeping' gene such as GAPDH. The primers generally used for reverse transcription are random hexamers or oligo dT (or both). Thus, it is clear that authors have not normalized the TERC transcripts to such a control. Could the authors please explain why this was not done?

We appreciate these comments. We have normalized the *TERC* transcripts to *U6* transcripts, an approach that is widely used for qPCR analysis of non-coding RNA quantification. We also used similar stem-loop backbone RT primer for *U6*; both RT primers were premixed before reverse transcription to improve quantification accuracy.

3. Possibly more importantly, the primer above used for the RT step is not present in the 451 nucleotide human *TERC* transcript which is:

GGGTTGCGGAGGGTGGGCCTGGGAGGGGTGGTGGCCATTTTTTGTCTAACCTAACT
GAGAAGGGCGTAGGCGCCGTGCTTTTGTCTCCCCGCGCGCTGTTTTTCTCGCTGACTTT
CAGCGGGCGGAAAAGCCTCGGCCTGCCGCCTTCCACCGTTCATTCTAGAGCAAACAA
AAAATGTCAGCTGCTGGCCCGTTCGCCCTCCCGGGGACCTGCGGGCGGGTTCGCCTGC
CCAGCCCCCGAACCCCGCCTGGAGGCCGCGGTTCGGCCCGGGGCTTCTCCGGAGGCAC
CCACTGCCACCGCGAAGAGTTGGGCTCTGTCAGCCGCGGGTCTCTCGGGGGCGAGGG
CGAGGTTTCAGGCCTTTCAGGCCGCAGGAAGAGGAACGGAGCGAGTCCCCGCGCGCG
GCGCGATTCCCTGAGCTGTGGGACGTGCACCCAGGACTCGGCTCACACATGC

Therefore, it is unclear how the PCR was successful if the incorrect RT primer was used. Could the authors please explain this?

The reviewer is directed to the responses to concerns 1 and 2 above.

On possibility is that the authors did not use DNase treatment, and subsequently did not include a 'no-reverse transcriptase' control in the RT reactions to assess DNA contamination in the PCR. If so, this would account for all of the fairly uniform qPCR results.

The reviewer is correct. To avoid contamination from genomic DNA, we pre-treated the RNA samples with DNase before reverse transcription, and we used a stem-loop RT primer for reverse transcription. When we establish this method, apart from the 'no-reverse transcriptase' control, we also used 'no-template' (no cDNA) samples in control qPCR assays.

4. While the forward primer used for TERC qPCR (TTCAGGCCTTTCAGGCCGCAGGAA) is present in this TERC sequence (underlined above) the reverse complement of the reverse primer used (TGGTGTCGTGGAGTCGGC) is not present in this sequence. Thus, it is not clear how the PCR worked for TERC? Did the authors sequence verify the PCR product??

As far as can tell, collectively these points appear to invalidate the qPCR results. Thus, an explanation addressing each of these would be necessary for me to recommend this manuscript to be taken further.

The concern regarding this primer was addressed above in our response to concern 2. We sequenced the PCR products to verify the amplification of TERC.

5. Could the authors please perform the TriFC experiment with the TERC C106 mutant? This would provide a conclusive result that the observed HuR interaction is mediated by C106 - as well as more fully validate the TriFC method to address this question.

As requested by the reviewer, we have performed TriFC analysis to test the influence of *TERC* C106 methylation on the association of HuR with *TERC* and have included these results in the revised Supplementary Information. As shown in Fig. S9, mutation of C106 did not seem to influence the association of HuR with *TERC*, in keeping with the results shown in Fig. S8.

6. Is the legend for Fig S4C correct?

Yes, the legend of Fig S4C is correct. We measured the levels of *TERC* in HeLa cells stably transfected with a vector expressing shHuR to test if long-term knockdown of HuR influenced the levels of *TERC*.

Reviewer #3 (Remarks to the Author):

As pointed out by other reviewers, this is a thorough evaluation of the effect of RNA binding protein HuR on regulation of telomerase. I have one major concern, regarding the authors response to Reviewer #3 on the number of GFP+ LSK HSC transplanted. The numbers of transplanted cells stated in the response, namely $4-6 \times 10^3$ or $1.2 - 1.8 \times 10^4$ seem very unrealistic to me, as this would correspond to the ENTIRE HSC population from the 4 long bones of at least 2-3 adult mice. More concerning is the purported number of LSK sorted, 1×10^5 , which correspond to about 20-25 mice worth of HSC. This would be an extremely time consuming and costly endeavor. Furthermore it would literally take days to FACS sort this many LSK HSC.

We agree with the reviewer that it would be very time-consuming to sort 1×10^5 LSK HSCs directly using FACS. Therefore, we used an anti-c-KIT antibody to enrich for HSCs by magnetic bead cell sorting (MACS) before FACS sorting. In addition, by crushing all the bones, including bones from hind legs, forelegs, pelvis, spine and sternum, we usually obtain 2×10^8 of total bone marrow cells from each mouse. The number of sorted LSK HSCs is about 1×10^5 (0.1% of BM cells) per mouse.

How many mice were used as LSK HSC donors in this experiment?

Four mice were used as LSK HSC donors in this experiment.

How long did it take to sort the LSK HSC?

It takes about 20-30 min to sort 1×10^5 LSK cells from the MACS-enriched c-KIT-positive BM cells.

The FACS data for sorting the LSK HSC must be shown, at least as a supplemental figure.

We have included representative FACS results for the sorting of LSK cells from the MACS-enriched c-KIT-positive BM cells (as shown in Fig. S13).

I cannot support publication of this paper until these 2 questions are answered, and I have seen the FACS data (dot plot).

We kindly direct the reviewer to the answers above and to the FACS data in the revised Supplementary Information (Fig. S13).

Reviewers' comments:

Reviewer #1 (Remarks to the Author):

The authors have added some mechanistic characterizations.

Reviewer #2 (Remarks to the Author):

I thank the authors for addressing each of my concerns.

Reviewer #3 (Remarks to the Author):

The authors have provided clarification regarding my concern about the LSK work. However, I still have concerns about the FACS sorting of LSK, even after the MACS enrichment, and the time claimed (20-30 minutes) to sort 1×10^5 LSK cells. What is the fold enrichment achieved using MACS c-Kit+ selection ? What was the flow rate (drops per second) used when sorting ? What was the make (manufacturer & model) of the FACS machine used ? It seems it should take hours, not minutes, to effectively sort 1×10^5 LSK cells.... I would also ask that the authors also include, as part of the supplementary material, a detailed description of the methods used to isolate the bone marrow and sort the LSK cells. References should be included, as needed.

Response to the reviewer 3's comments

Reviewer #3 (Remarks to the Author):

The authors have provided clarification regarding my concern about the LSK work. However, I still have concerns about the FACS sorting of LSK, even after the MACS enrichment, and the time claimed (20-30 minutes) to sort 1×10^5 LSK cells. What is the fold enrichment achieved using MACS c-Kit⁺ selection ? What was the flow rate (drops per second) used when sorting ? What was the make (manufacturer & model) of the FACS machine used ? It seems it should take hours, not minutes, to effectively sort 1×10^5 LSK cells....

I would also ask that the authors also include, as part of the supplementary material, a detailed description of the methods used to isolate the bone marrow and sort the LSK cells. References should be included, as needed.

We appreciate the comments from the reviewer. The fold of enrichment achieved using MACS based cKit⁺ selection is 20 fold. The flow rate (drops per second) was 8000 events per second. BD Influx Cell Sorter with 86 um nozzle was used for sorting.

After MACS based enrichment for cKit⁺ BM cells, the frequency of LSK cells was 2% in total events. With the flow rate of 8000 events per second, we usually could obtain 5000 LSK cells per minute. Therefore, it usually took 20-30 minutes to sort 1×10^5 LSK cells. Detailed description of the methods used to isolate the bone marrow and sort the LSK cells have been included in the supplementary Information.

References

1. Ema H, Morita Y, Nakauchi H, Matsuzaki Y. Isolation of murine hematopoietic stem cells and progenitor cells. *Current protocols in immunology*, **Chapter 22**, Unit 22B 21 (2005).
2. Frascoli, M., M. Proietti, and F. Grassi, Phenotypic analysis and isolation of murine hematopoietic stem cells and lineage-committed progenitors. *J Vis Exp* **65**, pii: 3736. doi: 10.3791/3736 (2012).
3. Ema, H., et al., Adult mouse hematopoietic stem cells: purification and single-cell assays. *Nat Protoc* **1**, 2979-87 (2006).

REVIEWERS' COMMENTS:

Reviewer #3 (Remarks to the Author):

The authors have satisfactorily addressed my concerns.